# Beyond Fixed Horizons: A Theoretical Framework for Adaptive Denoising Diffusions

## Abstract

We introduce a new class of adaptive denoising diffusion models (ADDMs) that, unlike conventional denoising diffusion models, achieve a time-homogeneous structure for both the noising and denoising processes, allowing the number of steps to adaptively adjust based on the noise level. This is accomplished by conditioning the forward process using Doob's $h$-transform, which terminates the process at a suitable sampling distribution at a random time. The model is particularly well suited for generating data with lower intrinsic dimensions, as the termination criterion simplifies to a first-hitting rule. A key feature of the model is its adaptability to the target data, enabling a variety of downstream tasks using a pre-trained unconditional generative model. These tasks include natural conditioning through appropriate initialisation of the denoising process and classification of noisy data. Several proof-of-concept experiments on the MNIST dataset are provided to demonstrate the applied potential of ADDMs.

## 1 Introduction

Denoising Diffusion Models (Ho et al., 2020; Song et al., 2021b) have gained significant attention in recent years due to their ability to generate high-quality data samples by iteratively denoising simple distributions based on the learned dynamics of a time-reversed forward noising process initalised in the target data distribution (Niu et al., 2020; Dhariwal & Nichol, 2021; Ho et al., 2022; Watson et al., 2023; Yang et al., 2023; Evans et al., 2024). A key limitation of these models, however, is their reliance on a fixed time horizon, which introduces an artificial time dependency in the drift function of the backward process. As a result, the generative denoising process follows a predefined number of steps, regardless of the actual level of noise present along the generated path.

To overcome this limitation, we introduce a novel class of adaptive denoising diffusion models (ADDMs) that dynamically adapt to the state of the denoising process. By replacing the fixed deterministic time horizon with a random one and conditioning the behaviour of the forward process at a terminal time, our approach achieves greater flexibility and state awareness. The foundation of our method lies in Doob's $h$-transforms with respect to underlying exponential times. While the theoretical groundwork for this concept exists, its explicit application and detailed exploration – particularly in comparison to deterministic time horizons – remains underrepresented in the literature.

A key feature of our model is its inherent adaptability: the number of denoising steps dynamically adjusts based on the noise level in the data, introducing a stochastic element. This randomness not only has the potential to enhance the generation process, but also allows denoising to start from partially noisy data, naturally incorporating conditioning. Moreover, the time required for denoising serves as an intuitive measure of the distance between noisy observations and the underlying data distribution, providing the basis for tasks such as classification and anomaly detection. The model's architecture also supports natural conditioning mechanisms, allowing seamless adaptation to diverse tasks without the need for task-specific design modifications.

Thanks to its flexible, time-homogeneous structure, our model offers a fresh perspective on generative tasks that enhances the adaptability and versatility of diffusion models and establishes a robust foundation for transfer learning.

**Contributions and structure**   Let us briefly summarize our main contributions and the structure of the paper: after discussing related work in Section 2, we first present theoretical results on the $h$-transform and time reversal for exponential time horizons in Section 3. Building on this foundation, we develop a universal and flexible diffusion model, accompanied by the associated learning theory, in Section 4. In this framework, we identify the *polarity* of the data distribution as a central assumption for successful learning, offering a new perspective on the manifold hypothesis. We discuss the adaptivity of the model across various application domains through transfer learning in Section 5 and provide experimental evidence for the performance of ADDMs for generation and the described down-stream tasks in Section 6.

## 2   Related work

This section reviews key works relevant to extending diffusion models beyond conventional time dependencies, incorporating elements such as random horizons, Doob's $h$-transform, and conditioning.

Ye et al. (2022) introduce first hitting diffusion models, which use first hitting times to capture the intrinsic geometry of data manifolds. Technically, their approach is related to ours in that it uses a random time horizon (in their case via first hitting times) and makes use of Doob's $h$-transform. However, their method defines a backward process that ensures that the generated data lies on a predetermined manifold. In contrast, our approach inverts this perspective and uses the forward process to create a more flexible generative framework that is not constrained to predefined manifolds. Instead, the data manifold is dynamically learned.

De Bortoli et al. (2021) explore diffusion generative modeling through the lens of Schrödinger Bridges (SB) and solving transport problems. Unlike traditional methods that require running forward SDEs over long durations, SB techniques generate samples in finite time. However, their approach is limited to a deterministic time horizon, where there are well-established connections between Schrödinger Bridges and $h$-transforms. Further developments in this direction are also presented in Peluchetti (2023) and Shi et al. (2023). In contrast, our framework accommodates random time horizons, allowing for both finite and long durations.

The paper Zhao et al. (2024) provides a comprehensive review of approaches to conditional sampling within generative diffusion models. It discusses methods that rely on joint distributions or pre-trained marginal distributions with explicit likelihoods to generate samples conditioned on certain information, addressing challenges in areas such as Bayesian inverse problems. In these approaches, the original unconditional processes are modified in various ways to introduce conditionality. Similarly, Didi et al. (2023) unifies conditional training and sampling within a common framework based on the $h$-transform. However, the reliance on a fixed time horizon leads to notable differences from the approach presented in our work.

Berner et al. (2024) introduce an optimal control perspective on denoising diffusion models, demonstrating how the backward generative process naturally emerges as a solution to a suitable optimal control problem. Proposition 4 establishes a similar connection in our model.

Denker et al. (2024) use $h$-transform techniques to introduce fine-tuning of a pre-trained diffusion model for conditional data generation. Additionally to the natural conditioning aspect that our model provides without second stage training, we demonstrate in Section 5.2 how an analogous procedure can be implemented for structured conditioning purposes in our modeling pipeline.

Finally, Christensen et al. (2025) builds on this paper and provides a detailed high-dimensional mathematical analysis of the natural conditioning property of time-reversals of exponentially killed Brownian motions as a particular sub-class of the general class of generative models that we introduce. This highlights the immediate usefulness of our model beyond pure unconditional data generation purposes.

# 3 Doob's $h$-transform and time-inversion from random times

Doob's $h$-transform is a versatile mathematical technique for adjusting the dynamics of a stochastic process. It formalizes the concept of conditioning the process on specific events occurring at a random time. This section outlines the key results relevant to our approach. For a more comprehensive discussion, including relevant literature and the proofs, readers are encouraged to consult the Appendix.

## 3.1 Doob's $h$-transform for random time horizons

In our approach, we will employ $h$-transforms of an underlying non-degenerate and symmetric $d$-dimensional diffusion process

$$dZ_t = b(Z_t)\, dt + \sigma(Z_t)\, dW_t,$$

Specifically, we make the following standard coefficient assumptions.

**Assumption 1.** *1. $b\colon \mathbb{R}^d \to \mathbb{R}^d$ and $\sigma\colon \mathbb{R}^d \times \mathbb{R}^{d\times d}$ are Lipschitz continuous.*

*2. $\Sigma := \sigma\sigma^\top$ is uniformly elliptic, that is there exists $\lambda > 0$ such that $\langle \xi, \Sigma(x)\xi\rangle \geq \lambda\|\xi\|^2$ for all $x, \xi \in \mathbb{R}^d$.*

Under these assumptions, there exists a strong solution $Z$, which is a strong Markov process with transition semigroup $(P_t)_{t\geq 0}$. While these regularity assumptions can be softened considerably, they are sufficiently general for the purposes of this paper. Additionally we will assume that the diffusion is symmetric (or self-dual) in the following sense:

**Assumption 2.** *1. The transition semigroup is symmetric wrt. a $\sigma$-finite reference measure $m$, that is $P_t(x, dy)\, m(dx) = P_t(y, dx)\, m(dy)$.*

*2. For every $t > 0$ we have $P_t(x, dy) = p_t(x, y)\, m(dy)$ for a measurable function $p_t$, called the transition density at time $t$ (wrt. $m$).*

Given these assumptions, the $m$-transition densities are symmetric, i.e. $p_t(x, y) = p_t(y, x)$. If $Z$ is recurrent, $m$ can be chosen as a $\sigma$-finite invariant measure (that is $m = mP_t$, where $mP_t(dy) := \int_{\mathbb{R}^d} P_t(x, dy)\, m(dx)$) and continuous transition densities $p_t(x, y)$ wrt. $m$ exist see e.g. Stroock & Varadhan (2006). Typically, $Z$ will be a Brownian motion (in this case $m$ is the Lebesgue measure) or a symmetric Ornstein–Uhlenbeck process on $\mathbb{R}^d$ (then, $\sigma = \mathrm{Id}$ and $b(x) = -Ax$ for a symmetric drift matrix $A$, which, if positive definite, yields a Gaussian $\mathcal{N}(0, \frac{1}{2}A^{-1})$ stationary law), but other self-dual processes are also possible. We impose Assumptions 1 and 2 for the remainder of the paper. The unkilled version of $Z$ then has the symmetric $r$-Green kernel

$$G_r(x, y) := \int_0^\infty \mathrm{e}^{-rt} p_t(x, y)\, dt, \quad r > 0, x, y \in \mathbb{R}^d.$$

Up to scaling, $G_r$ describes the pdf of $Z$ at an independent exponential time. For a Brownian motion, for example, $G_r$ can be found explicitly, see Remark 11.

To specify the idea of conditioning at a random time, we introduce two major modifications for $Z$:

1. The drift can be modified to attract the process towards desired states.

2. A random stopping time $\zeta$, referred to as the *lifetime*, is introduced to terminate the process at the point where the desired distribution is attained.

Both modifications are seamlessly implemented using the $h$-transform. The $h$-transform is constructed using an $r$-excessive function $h$, which can be defined via a probability measure $\beta$ on the state space $\mathbb{R}^d$ and a reference state $x_0 \in \mathbb{R}^d$. To condition the process $Z$ to be distributed according to $\beta$ at its killing time $\zeta$ when initiated from $x_0$, we define

$$h(x) := \int \frac{G_r(x, y)}{G_r(x_0, y)}\, \beta(dy) = \int G_r(x, y)\, \kappa(dy),$$

where $\kappa(dy) = \kappa_{x_0,\beta}(dy) = \frac{1}{G_r(x_0,y)}\beta(dy)$ is called *representing measure* of $h$. Based on this, the $h$-transformed process $Z^h$ is now a Markov process defined by the transition kernel

$$\mathbb{P}_x(Z_t^h \in dy) = \mathbb{E}_x\left[\frac{\mathrm{e}^{-rt}h(Z_t)}{h(x)}\mathbf{1}_{\{Z_t \in dy\}}\right].$$

The essential properties of $Z^h$ are as follows:

**Proposition 3.**     *1. $Z^h$ is killed at some random time $\zeta$, the lifetime, which is an a.s. finite random variable.*

    *2. Outside the support of $\beta$, $Z^h$ is an Itô diffusion with dynamics*

$$dZ_t^h = b^h(Z_t^h)\,dt + \sigma(Z_t^h)\,dW_t,$$
$$b^h(y) = b(y) + \sigma(y)\sigma(y)^\top\nabla\log h(y).$$

    *3. The distribution of $Z^h$ at its lifetime is supported in the support of $\beta$ and given by*

$$\mathbb{P}_x(Z_{\zeta-}^h \in dy) = \frac{G_r(x,y)}{h(x)}\kappa(dy).$$

*In particular,*

$$\mathbb{P}_{x_0}(Z_{\zeta-}^h \in dy) = \beta(dy).$$

The proof is deferred to Appendix A. This result demonstrates that outside the target data support, the guided process behaves like a standard diffusion but is pulled toward the target by the force $\nabla\log h$, where the process is terminated in finite time. The terminal dsitribution then matches exactly $\beta$ when the process is started in the reference state $x_0$ and otherwise is a weighted version thereof with weights depending on the initial state $x$.

## 3.2   Optimality properties of the $h$-transform

The $h$-transform is characterised by inherent optimality properties which underline its role as a canonical approach. We now demonstrate that it emerges as the solution to a stochastic stopping and control problem. In this context, the running cost component can be interpreted as minimizing the expected lifetime of the process while penalizing large drifts. By framing the $h$-transform within this stochastic control problem, we establish a connection to the KL divergence.

**Proposition 4.** *We write $k(u) = r + \frac{1}{2}\|u\|^2$, $g(y) = -\log h(y)$. For an (admissible) control $u$ consider the controlled process*

$$dZ_t^u = (b(Z_t^u) + \sigma(Z_t^u)u_t)\,dt + \sigma(Z_t^u)\,dW_t,\ t < \tau_\beta,$$

*where $\tau_\beta = \tau_\beta^u$ denotes the first entrance time into the support of $\beta$. The stochastic stopping and control problem of minimizing*

$$J(u,\tau,x) := \mathbb{E}_x\left[\int_0^\tau k(u_t)\,dt + g(Z_{\tau-}^u)\right], \qquad v(x) = \inf_{u,\ \tau \le \tau_\beta^u} J(u,\tau,x) \tag{1}$$

*in both $u$ and $\tau \le \tau_\beta$ is solved as follows:*

    *1. $v(x) = g(x) = -\log h(x)$.*

    *2. All pairs $(u^*,\tau)$ for the Markov feedback control $u^*$ given by*

$$u^*(x) = -\sigma(x)^\top\nabla g(x),$$

    *and arbitrary stopping times $\tau \le \tau_\beta$ are optimal.*

3. *For any admissible u,*

$$J(u, \tau, x) - v(x) = \mathrm{KL}\left(\left.\mathbb{P}_x^{Z^u}\right|_{\mathcal{F}_{\tau_\beta}} \middle\| \left.\mathbb{P}_x^{Z^h}\right|_{\mathcal{F}_{\tau_\beta}}\right),$$

*so the Kullback–Leibler divergence between the controlled law and the h-transformed law (up to $\tau_\beta$) equals the variational gap.*

The proof can be found in Appendix B. We note here that for the cases we are interested in later, the first entrance time into the support just happens to coincide with the lifetime $\zeta$.

### 3.3 Connection to time-inversion

Compared to the deterministic case, the time reversal from a random time is much less elaborated in the literature. In terms of the *h*-transform, however, the results are very clear: for the *h*-transformed process $Z^h$ with (finite) lifetime $\zeta$, we consider the time-reversed process

$$\overleftarrow{Z_s^h} := Z_{\zeta-s}^h, \, 0 < s < \zeta,$$

killed at $s = \zeta$. Let also $\alpha$ be a fixed initial distribution of $Z^h$, and define

$$\overleftarrow{h}(x) = \int \frac{G_r(x, y)}{h(y)} \, \alpha(dy). \tag{2}$$

We then have the following key result, whose proof is given in Appendix A.

**Proposition 5.** *1. $\overleftarrow{Z^h}$ has the same distribution as $Z^{\overleftarrow{h}}$ with initialisation $Z_0^{\overleftarrow{h}} \sim Z_{\zeta-}^h$; in particular,*

$$d\overleftarrow{Z_s^h} = b^{\overleftarrow{h}}(\overleftarrow{Z_s^h}) \, ds + \sigma(\overleftarrow{Z_s^h}) \, d\overleftarrow{W}_s,$$

$$b^{\overleftarrow{h}}(y) = b(y) + \sigma(y)\sigma(y)^\top \nabla \log \overleftarrow{h}(y),$$

*$\overleftarrow{W}$ a Brownian motion, outside the support of $\alpha$.*

*2. $\overleftarrow{Z^h}$ is killed on the support of $\alpha$.*

*3. When started in $x \notin \mathrm{supp}\,\alpha$, the distribution of $Z^{\overleftarrow{h}}$ at its lifetime $\zeta$ is given by*

$$\mathbb{P}_x(Z_{\zeta-}^{\overleftarrow{h}} \in dy) = \frac{G_r(x, y)}{\overleftarrow{h}(x)h(y)} \, \alpha(dy).$$

*In particular, for $\mathbb{P}_x(Z_{\zeta-}^h \in \cdot)$-a.e. $x \notin \mathrm{supp}\,\alpha$ it holds for the time reversed process $\overleftarrow{Z^h}$ that*

$$\mathbb{P}_x(\overleftarrow{Z_{\zeta-}^h} \in dy) = \frac{G_r(x, y)}{\overleftarrow{h}(x)h(y)} \, \alpha(dy).$$

The interpretation of the time reversal in terms of the *h*-transform has interesting consequences. Let us recall the concept of a polar set: for $Z^h$, $A \subseteq \mathbb{R}^d$ is called *polar* if $\mathbb{P}_x(Z_t^h \in A$ for some $t > 0) = 0$ for all $x \in \mathbb{R}^d$. Typical examples of polar sets for diffusions are sets on low-dimensional submanifolds. For instance, if $Z^h$ is a $d$-dimensional Brownian motion, any set with Hausdorff dimension less than $d - 2$ is polar, see e.g. Mörters & Peres (2010); a related result for general diffusions is also provided in Ramasubramanian (1988).

**Proposition 6.** *Assume that the support of the initial distribution $\alpha$ is polar for $Z^h$. Then, $\overleftarrow{Z^h}$ is killed at the first entry into the support of $\alpha$. In particular, the distribution of $\overleftarrow{Z^h}$ is entirely characterised by the drift $b^{\overleftarrow{h}}$ and the support of $\alpha$.*

*Proof.* The assumption implies that the starting distribution of $Z^h$ lies in a polar set, which is not visited during the rest of the lifetime. This means that the time-reversed process $\overleftarrow{Z^h}$ only visits this set at the time of killing. □

Given polarity of the support the termination criterion therefore simplifies considerably to a first hitting rule but since the true data distribution and hence also its support is not known in practice, we cannot check directly if the backward process has hit the target support. However, in Section 4.4 we demonstrate how Proposition 6 can be exploited to obtain an equivalent characterisation of the first hitting time of the support as a first explosion time of the score, thus making score estimation the only necessary requirement for implementing the model. There, we also discuss implications for our model, when the polarity hypothesis is not satisfied.

### 3.4 Key examples

1. In the case $\beta(dy) = rG_r(x_0, y) m(dy)$ (the distribution of $Z$ at the exponential time when started in $x_0$), we have

$$h(x) = \int \frac{G_r(x, y)}{G_r(x_0, y)} \beta(dy) = \int rG_r(x, y) m(dy) = 1.$$

   Then, $Z^h$ has the same dynamics as $Z$ and $\zeta$ is an independent $\text{Exp}(r)$-time. The time-reversed dynamics are determined by $\overleftarrow{h}(x) = \int G_r(x, y) \alpha(dy)$.

2. In the case $\beta(dy) = \delta_{x_1}$ for some $x_1 \in \mathbb{R}^d$, we have

$$h(x) = G_r(x, x_1)/c_{x_1}, \ c_{x_1} = G_r(x_0, x_1).$$

   $h$ is thus independent of the starting point, except for a multiplicative constant that is irrelevant for the $h$-transform, so we may choose $c_{x_1} = 1$ w.l.o.g. $Z^h$ is thus an exponential bridge killed in $x_1$, no matter where the starting point was, with drift

$$b_h(x) = b(x) + \sigma(x)\sigma(x)^\top \nabla \log G_r(x, x_1).$$

   In this case,

$$\overleftarrow{h}(x) = \int \frac{G_r(x, y)}{G_r(x_1, y)} \alpha(dy).$$

3. In case $Z = W$ is a Brownian motion, we consider radially symmetric functions $h(x) = f(|x|)$ of the form

$$h(x) = \int_{\partial B_R} G_r(x, y) \sigma_R(dy),$$

   where $\sigma_R$ denotes the surface measure on the sphere with radius $R$. We obtain that (except for scaling)

$$f(y) = y^{-\nu} I_\nu(y\sqrt{2r}), \ \ \nu = \frac{d-2}{2},$$

   $I_\nu$ is a modified Bessel function of the first kind, and this results in the forward process $Z^h$ with drift

$$b^h(x) = \nabla \log h(x) = \sqrt{2r} \frac{I_{\nu+1}(\|x\|\sqrt{2r})}{I_\nu(\|x\|\sqrt{2r})} \frac{x}{\|x\|},$$

   killed when exiting $B_R$ (in the sense of a last exit). For large dimensions $d$ and moderate values of $\|x\|$, it holds that $b^h(x) \approx \frac{2r}{d} x$ (see 10.41 in DLMF), so that the forward process $Z^h$ can be approximated using the (non-stationary) Ornstein–Uhlenbeck process dynamics

$$d\widetilde{Z}_t = \frac{2r}{d} \widetilde{Z}_t \, dt + dW_t. \tag{3}$$

## 4 The proposed model

Standard diffusion generative models rely on a deterministic time horizon, leading to time-dependent backward processes and inefficiencies in the forward process due to excessive noise application. Two promising directions for improving these models can be identified: (1) replacing the deterministic time horizon with a randomised one to achieve time-homogeneous backward dynamics, and (2) introducing conditioning in the forward process to reduce the need for extensive noise application.

We achieve the simultaneous implementation of both modifications in a unified framework employing the $h$-transform described above. To this end, we use an appropriate choice of the process $Z^h$ as our forward process and $\overleftarrow{Z^h}$ as backward process, as discussed next.

### 4.1 Three possible implementations for unconditional sampling

The examples from Section 3.4 immediately suggest possible realisations of our framework. Specifically, we use $Z^h$ as a forward process initialised in the data distribution $\alpha$ to learn the drift of the backward process $\overleftarrow{Z^h}$ (see the following sections for details). To generate (unconditional) samples from $\alpha$, we require suitable initial distributions for $\overleftarrow{Z^h}$, which are (approximately) given in the following examples:

1. **Long exponential time horizons:** If we set $h \equiv 1$, then $Z^h$ has the same dynamics as the reference process $Z$ and $\zeta$ follow an $\mathrm{Exp}(r)$ distribution, where for smaller $r$ we have longer average lifetimes. If $Z$ is ergodic (say, an ergodic Ornstein–Uhlenbeck process), then $\mathbb{P}_\alpha(Z_\zeta \in dy)$ is close to the stationary distribution of $Z$ and we may initialise the backward process $\overleftarrow{Z^h}$ in the stationary distribution of $Z$. This is the natural analog to variance preserving (VP) diffusion models (Song et al., 2021b) but with an exponentially distributed forward run-time $\zeta$ instead of a fixed time $T$. If $Z$ is a Brownian motion and thus non-ergodic, then conditional on $Z_0 = x$, $Z_\zeta \sim \mathrm{Laplace}(x, r^{-1}I_d)$. Thus, similarly to variance exploding (VE) diffusion models with fixed sampling time (Song et al., 2021b), for sufficiently small exponential killing parameter $r$ we can initialise the reverse SDE in a high variance $\mathrm{Laplace}(0, r^{-1}I_d)$ distribution.

2. **Exponential bridge:** In the second example, the forward process $Z^h$ is an exponential bridge targeting a specific state $x_1$. Here, we can sample from the data distribution by starting the backward process $\overleftarrow{Z^h}$ at $Z^h_{\zeta-} = x_1$.

3. **Hitting a large sphere:** In the third example, if $\alpha$ is compactly supported and the radius $R$ is chosen sufficiently large, $\mathbb{P}_\alpha(Z^h_{\zeta-} \in dy)$ is approximately a uniform distribution over the sphere, which can then serve as an initial distribution for $\overleftarrow{Z^h}$. If helpful, the forward process can be approximated by (3), which allows a more direct sampling.

In the idealised setting where $\alpha$ is known analytically, generation via the backward process for unconditional sampling, i.e., sampling from $\alpha$, proceeds as described in Algorithm 1. This algorithm is derived from the results presented in Section 3.3. The simulation of the backward process up to the lifetime $\zeta$ can be implemented using standard numerical methods, such as the Euler–Maruyama scheme. In the most general setup, the backward process terminates at a randomised Markovian stopping time of the form

$$\zeta = \inf\{t : A^h_t \geq E\},$$

where $E$ is an independent exponential random variable and $A^h_t$ is an additive functional supported within the data distribution, which describes the measure of killing. For a detailed mathematical discussion and numerical approximations of $A^h$, see Bally (1989); Stoica (1992); Christensen & Schultz (2024). This framework directly enables simulation but poses challenges for estimation. However, the situation becomes significantly simpler when the data distribution $\alpha$ resides within a polar set, as will be discussed in the following section.

---

**Algorithm 1** Idealised generation when no learning is necessary

---

**Input:** $r > 0$, Green kernel $G_r$ of diffusion $Z$, forward transform $h$, target distribution $\alpha$, backward initial distribution $\beta' \approx \mathbb{P}_\alpha(Z^h_{\zeta-})$

Set $\overleftarrow{h}(x) = \int \frac{G_r(x,y)}{h(y)} \alpha(dy)$

**Generation:**

draw $x \sim \beta'$

Simulate path $(y_t)_{t \in [0,\zeta)}$ of $h$-transform $Y = \overleftarrow{Z^h}$ initialised in $x$ until lifetime $\zeta$, with dynamics given in Proposition 5;

**Output:** $y_{\zeta-}$

---

### 4.2 Polarity hypothesis

The target distribution $\alpha$ is typically unknown in practice. Consequently, we lack direct access to $\overleftarrow{h}$ as defined in (2), which is required to employ the backward generating process $Y = \overleftarrow{Z^h}$. Thus, $\overleftarrow{h}$ must be inferred from the data. As discussed in Section 3, both the drift of the backward process and the mechanism governing killing must be learned. While the killing mechanism can be described by a measure on the state space $\mathbb{R}^d$ (the killing measure), this adds significant complexity to the learning process.

A natural approach is to terminate the backward process as soon as a meaningful element of the data distribution is encountered. Proposition 6 offers a criterion for when this is feasible: specifically, when the data distribution is concentrated in a polar set for the forward process, such as a lower-dimensional manifold. This aligns naturally with the manifold hypothesis: this well-explored concept in the literature assumes that high-dimensional data typically lie on or near a low-dimensional manifold (or a union of such) (Loaiza-Ganem et al., 2024) and has been empirically verified for image data (Pope et al., 2021; Brown et al., 2023).

In our framework, we adopt a slightly more general assumption, referred to as the *polarity hypothesis*, which posits that the data resides in a polar set. This generalisation allows us to disentangle learning of the drift and the killing mechanism, as detailed in the following sections.

### 4.3 Learning the drift

Here we focus on the case where the polarity hypothesis is satisfied, such that by Proposition 6 the lifetime of $\overleftarrow{Z^h}$ is given by the first hitting time of $\Omega := \mathrm{supp}\,\alpha$, which we assume to be known for the moment. Based on Proposition 5, we aim to fit the data to a class of time-homogeneous diffusion processes

$$dY^\theta_s = \overleftarrow{b^\theta}(Y^\theta_s)\,ds + \sigma(Y^\theta_s)\,dW_s, \quad Y^\theta_0 \sim \mathbb{P}_\alpha(Z^h_{\zeta-} \in \cdot),$$

$$\overleftarrow{b^\theta}(y) = b(y) + \sigma(y)\sigma(y)^\top s_\theta(y),$$

induced by a suitable class of candidate functions $\mathcal{S} = \{s_\theta : \theta \in \Theta\}$, and for an optimization objective and corresponding optimiser $\theta^*$ to be determined below, run the backward process $Y^{\theta^*}$ until an appropriate stopping time. Ideally, we would like to stop in the first hitting time of $\Omega$ by $Y^{\theta^*}$, i.e., in $\zeta^{\theta^*} = \tau_\Omega(Y^{\theta^*})$, denoting $\tau_A(X) := \inf\{t : X_t \in A\}$ for a process $X$ and a set $A \subset \mathbb{R}^d$. However, in the typical situation where $\Omega$ is a lower dimensional manifold with no simple structure, designing a candidate class $\mathcal{S}$ that guarantees $\mathbb{P}(\zeta^{\theta^*} < \infty) = 1$ is a difficult and highly problem-specific task (e.g., a $C^2$ submanifold $\Omega$ is polar for non-degenerate diffusions if its Hausdorff dimension is no larger than $d-2$ (Friedman, 1975, Chapter 11), (Ramasubramanian, 1988)). Instead, for some small $\varepsilon > 0$, we consider the closed $\varepsilon$-environment $\Omega_\varepsilon := \{x \in \mathbb{R}^d : d(x, \Omega) \leq \varepsilon\}$ and run $Y^{\theta^*}$ until

$$\zeta^{\theta^*}_\varepsilon = \tau_{\Omega_\varepsilon}(Y^{\theta^*}).$$

Thus, we do not target $\alpha$ directly, but the distribution of $\overleftarrow{Z^h}_{\zeta_\varepsilon}$ for

$$\zeta_\varepsilon := \tau_{\Omega_\varepsilon}(\overleftarrow{Z^h}).$$

This is justified for small $\varepsilon > 0$, since by continuity of the sample paths, we have that $\tau_{\Omega_\varepsilon}(\overleftarrow{Z}^h)$ almost surely increases to $\zeta$ as $\varepsilon \downarrow 0$ if $\Omega$ is closed, and therefore obtain the a.s. convergence $\lim_{\varepsilon \downarrow 0} \overleftarrow{Z}^h_{\zeta_\varepsilon} = \overleftarrow{Z}^h_{\zeta-} \sim \alpha$. This is comparable to early stopping of the generating process in standard diffusion models with deterministic time horizon, which implies that the generative model targets the original data set blurred by some small Gaussian noise. On a forward time scale, the first entrance time in $\Omega^\varepsilon$ of the backward process $\overleftarrow{Z}^h$ corresponds to the last exit time $\sigma_\varepsilon$ of $\Omega_\varepsilon$ by the forward process $Z^h$, that is,

$$\sigma_\varepsilon := \sup\{t < \zeta : Z^h_t \in \Omega_\varepsilon\} = \zeta - \zeta_\varepsilon.$$

As a natural optimisation objective, we therefore target the Kullback–Leibler divergence between the law $\overleftarrow{\mathbb{P}}^\varepsilon$ of the theoretical generating process $\overleftarrow{Z}^h$ killed in $\zeta_\varepsilon$ and the law $\mathbb{P}^{\theta,\varepsilon}$ of the parametrised generating process $Y^\theta$ killed at first entrance into $\Omega_\varepsilon$, i.e., at $\zeta^\theta_\varepsilon := \tau_{\Omega_\varepsilon}(Y^\theta)$. To this end, we set

$$\theta^* \in \operatorname*{arg\,min}_{\theta \in \mathcal{S}} \mathcal{L}_{\mathrm{ex}}(\theta),$$

$$\mathcal{L}_{\mathrm{ex}}(\theta) := \mathbb{E}\Big[ \int_0^{\zeta_\varepsilon} \|s_\theta(\overleftarrow{Z}^h_t) - \nabla \log \overleftarrow{h}(\overleftarrow{Z}^h_t)\|^2 \, dt \Big]$$

$$= \mathbb{E}\Big[ \int_{\sigma_\varepsilon}^{\zeta} \|s_\theta(Z^h_t) - \nabla \log \overleftarrow{h}(Z^h_t)\|^2 \, dt \Big]$$

which is the natural analog to the *explicit score matching* objective in standard diffusion models. Then, given sufficient integrability conditions, Girsanov's theorem indeed yields that

$$\mathrm{KL}\big(\overleftarrow{\mathbb{P}}^\varepsilon \,\|\, \mathbb{P}^{\theta,\varepsilon}\big) = \frac{1}{2}\mathbb{E}\Big[ \int_0^{\zeta_\varepsilon} \|\sigma^{-1}(\overleftarrow{Z}^h_t)(b^{\overleftarrow{h}}(\overleftarrow{Z}^h_t) - b^{\overleftarrow{\theta}}(\overleftarrow{Z}^h_t))\|^2 \, dt \Big]$$

$$\leq \frac{C_\sigma}{2}\mathcal{L}_{\mathrm{ex}}(\theta),$$

for $C_\sigma := \sup_{x \in \mathbb{R}^d} \|\sigma^{-1}(x)\|^2$. Consequently,

$$\mathrm{KL}\big(\mathbb{P}(\overleftarrow{Z}^h_{\zeta_\varepsilon} \in \cdot) \,\|\, \mathbb{P}(Y^{\theta^*}_{\zeta^{\theta^*}_\varepsilon} \in \cdot)\big) \lesssim \min_{\theta \in \mathcal{S}} \mathbb{E}\Big[ \int_0^{\zeta_\varepsilon} \|s_\theta(\overleftarrow{Z}^h_t) - \nabla \log \overleftarrow{h}(\overleftarrow{Z}^h_t)\|^2 \, dt \Big],$$

showing that if $\mathcal{S}$ is rich enough to guarantee a high approximation quality of $\nabla \log \overleftarrow{h}$ on $(\Omega_\varepsilon)^c$ (e.g., a suitable class of neural networks), our generative process produces high quality samples for slightly blurred target data. However, the general inaccessibility of $\overleftarrow{h}$, and thus of $\mathcal{L}_{\mathrm{ex}}$, requires us to determine a tractable training objective that is comparable to $\mathcal{L}_{\mathrm{ex}}$. This is provided by our next result, which in our proposed model can be understood as an analog of the *denoising score matching loss* (Vincent, 2011) employed in standard diffusion models.

**Proposition 7.** *Assume that* $\mathbb{E}\Big[ \int_0^\zeta |\langle \nabla_y \log \overleftarrow{h}(Z^h_t), s_\theta(Z^h_t) \rangle| \, dt \Big] < \infty$ *and that* $\nabla_y \int \frac{G_r(y,z)}{h(z)} \alpha(dz) = \int \nabla_y \frac{G_r(y,z)}{h(z)} \alpha(dz)$. *Then, there exists a constant $C$ independent of $\theta$, such that*

$$\mathcal{L}_{\mathrm{ex}}(\theta) = \mathcal{L}(\theta) - C_\varepsilon(\theta) + C,$$

*where*

$$\mathcal{L}(\theta) := \mathbb{E}\Big[ \int_{\sigma_\varepsilon}^{\zeta} \|s_\theta(Z^h_t) - \nabla_{Z^h_t} \log G_r(Z^h_t, Z^h_0)\|^2 \, dt \Big]$$

*and*

$$C_\varepsilon(\theta) := 2\mathbb{E}\Big[ \int_0^{\sigma_\varepsilon} \big\langle s_\theta(Z^h_t), \nabla_{Z^h_t} \log \frac{G_r(Z^h_t, Z^h_0)}{\overleftarrow{h}(Z^h_t)} \big\rangle \, dt \Big].$$

Since we assumed $\Omega$ to be polar, $C_\varepsilon(\theta)$ vanishes as $\varepsilon \to 0$, so that for small $\varepsilon > 0$ we expect minimisers of $\mathcal{L}$ – which is accessible for a given Green kernel $G_r$ and can be efficiently approximated by a Monte-Carlo

---

**Algorithm 2** Generation for unknown $\overleftarrow{h}$ and known polar data support $\Omega$

---

**Input:** data $\{y_i\}_{i=1}^n \overset{iid}{\sim} \alpha$, $N \in \mathbb{N}$, $\varepsilon > 0, r > 0$, Green kernel $G_r$ of diffusion $Z$, enlarged data support $\Omega_\varepsilon$, forward transform $h$, backward initialisation $\beta' \approx \mathbb{P}_\alpha(Z_{\zeta_-}^h \in \cdot)$ with $\operatorname{supp}\beta' \subset \Omega_\varepsilon^c$, function class $\mathcal{S}$.
**for** $j = 1$ **to** $N$ **do**
    draw $y_{i_j}$ with replacement from $\{y_i\}_{i=1}^n$ and simulate path $(z_t^{h,i_j})_{t \in [0,\zeta^{i_j})}$ of $h$-transform $Z^h$ started in $y_{i_j}$;
**end for**
**Training:** Learn $\widehat{\theta} = \arg\min_{\theta \in \mathcal{S}} \widehat{\mathcal{L}}(\theta)$ for

$$\widehat{\mathcal{L}}(\theta) = \frac{1}{N}\sum_{j=1}^N \int_{\sigma_\varepsilon^{i_j}}^{\zeta^{i_j}} \|s_\theta(z_t^{h,i_j}) - \nabla_1 \log G_r(z_t^{h,i_j}, y_{i_j})\|^2 \, dt$$

**Generation:**
Draw $x$ from $\beta'$
Simulate path $(y_t^{\widehat{\theta}})_{t \in [0,\tau_{\Omega_\varepsilon}(y^{\widehat{\theta}})]}$ of $Y^{\widehat{\theta}}$ initialised in $x$
**Output:** $\widehat{\theta}, y_{\tau_{\Omega_\varepsilon}(y^{\widehat{\theta}})}^{\widehat{\theta}}$

---

estimator $\widehat{\mathcal{L}}$ based on simulated forward trajectories of $Z^h$ – to be also approximate minimisers of $\mathcal{L}_{\mathrm{ex}}$. This motivates the generative Algorithm 2, which outputs an estimated drift parameter $\widehat{\theta}$ and a corresponding sample from the estimated backward generative process $Y^{\widehat{\theta}}$ initialised in a distribution $\beta'$ approximating the true forward terminal distribution $\mathbb{P}_\alpha(Z_{\zeta_-}^h \in \cdot)$. We note once again that a significant difference from most other methods is that there is no time dependence to be learned. This is also a feature of the first hitting model from Ye et al. (2022), which, however, relies on non-polarity of $\Omega$ and access to the $\Omega$-dependent Poisson kernel $\mathbb{P}_x(Z_{\tau_\Omega} \in dz)$ for training and generation purposes. Usually, this is analytically tractable only for simple sets $\Omega$. In contrast, our learning and generation step only requires access to the Green kernel $G_r$ of the *unconditioned* forward process $Z$, which is independent of $\Omega$ and therefore allows for enhanced flexibility. Furthermore, successful implementations of standard denoising diffusion models rely on a temporal weighting of the denoising score loss objective based on the noise schedule to normalise the signal strength. This can be naturally introduced into our denoising score loss as well, e.g. by using the *spatial* weighting factor $\|\nabla_{Z_t^h} \log G_r(Z_t^h, Z_0^h)\|^{-2}$. Further details on such practical aspects are discussed in detail in Section 6.

### 4.4 (Implicit) manifold learning

We now turn to the most general setting, where the polar data support $\Omega = \operatorname{supp}\alpha$ is unknown. Since the generative mechanism proposed in Algorithm 2 fundamentally requires $\Omega_\varepsilon$ as input in order to know where to stop the generative process, we must infer $\Omega_\varepsilon$ from the data. To this end, we discuss two different strategies: first, a traditional statistical plug-in approach that simply replaces $\Omega_\varepsilon$ in Algorithm 1 with a separately obtained estimate $\widehat{\Omega_\varepsilon}$. The second approach is based on the observation that under the polarity hypothesis, the unknown drift component $\nabla \log \overleftarrow{h}$ implicitly encodes the geometry of the data manifold $\Omega$, cf. Proposition 10. This motivates an adaptive stopping criterion that only requires an estimate of $\nabla \log \overleftarrow{h}$ as input and may be interpreted as a first hitting time of an implicit manifold estimate.

**Approach based on separation of manifold and drift estimation** A first—but perhaps naïve— strategy is to build an estimator $\widehat{\Omega_\varepsilon}$ based on the given data in a pre-processing step, and then use a plug-in approach to draw samples based on Algorithm 2. This is formalised in Algorithm 3.

To construct $\widehat{\Omega_\varepsilon}$ given a data sample $(Y_i)_{i=1}^n \overset{iid}{\sim} \alpha$, we can consider the following two natural approaches:

---

**Algorithm 3** Full "naïve" generative algorithm

---

**Input:** data $\{y_i\}_{i=1}^n \overset{iid}{\sim} \alpha$, $\varepsilon > 0, r > 0$, Green kernel $G_r$ of diffusion $Z$, estimator $\widehat{\Omega_\varepsilon}$ of enlarged data support, forward transform $h$, backward initial distribution $\beta' \approx \mathbb{P}_\alpha(Z_{\zeta-}^h)$ with $\operatorname{supp} \beta' \subset \widehat{\Omega_\varepsilon}^c$, function class $\mathcal{S}$.

Set $\Omega_\varepsilon \leftarrow \widehat{\Omega_\varepsilon}$

**Generation:** Run Algorithm 2

**Output:** $\widehat{\theta}, y_{\tau_{\widehat{\Omega_\varepsilon}}(y^{\widehat{\theta}})}^{\widehat{\theta}}$

---

First, we may target $\Omega_\varepsilon$ directly by blurring the data via $Y_i^\varepsilon = Y_i + \eta_i^\varepsilon$, where $(\eta_i^\varepsilon)_{i=1}^n$ is some iid noise that is independent of the data sample and absolutely continuous with support $B(0, \varepsilon)$, e.g., $\eta_i^\varepsilon \sim \mathcal{U}(B(0, \varepsilon))$. Then, $Y_i^\varepsilon$ is absolutely continuous with support $\Omega_\varepsilon$ and density $\pi^\varepsilon(x) = \int_\Omega \pi_{\eta^\varepsilon}(x - y)\,\alpha(dy)$. Recovering the compact support $\Omega_\varepsilon$ of such a data sample is a well-studied statistical problem. The perhaps most common approaches are plug-in estimators Cuevas & Fraiman (1997) that set

$$\widehat{\Omega_\varepsilon} = \{\widehat{\pi^\varepsilon} > \delta_n\}$$

for some nonparametric estimator $\widehat{\pi^\varepsilon}$ of $\pi^\varepsilon$ and a tuning parameter $\delta_n \to 0$, or the simple and intuitive *Devroye–Wise* estimator Devroye & Wise (1980) given by

$$\widehat{\Omega_\varepsilon} = \bigcup_{i=1}^n B(Y_i^\varepsilon, \delta_n).$$

Minimax optimality of these estimators with respect to metrics such as the Hausdorff metric or the symmetric difference volume has been established under different assumptions on the data density $\pi^\varepsilon$ and the geometry of $\Omega_\varepsilon$ (Korostelëv & Tsybakov, 1993; Mammen & Tsybakov, 1995; Cuevas & Fraiman, 1997; Cuevas & Rodríguez-Casal, 2004; Biau et al., 2008). The rates, however, generally slow down exponentially in terms of the ambient data dimension $d$, which motivates the second approach that allows to exploit directly the lower-dimensional structure of $\Omega$.

For this second approach, instead of estimating $\Omega_\varepsilon$ directly, we may first construct an estimator $\widehat{\Omega}$ of the true data support $\Omega$ based on the unmodified data sample $(Y_i)_{i=1}^n$ and then set

$$\widehat{\Omega_\varepsilon} = (\widehat{\Omega})_\varepsilon.$$

This approach might be more promising for high-dimensional data since a lot of progress has been made in recent years on statistical theory and algorithmic implementation for support estimation of distributions concentrated on lower-dimensional manifolds. Important contributions that provide estimators $\widehat{\Omega}$ with provable convergence rates only depending on the smoothness $s$ and the dimension $d' < d$ of the data manifold include Genovese et al. (2012); Ma & Fu (2012); Kim & Zhou (2015); Aamari & Levrard (2018; 2019); Aamari et al. (2023). While computationally more involved than the simple estimators based on noised data discussed above, the estimators from Aamari & Levrard (2018; 2019); Aamari et al. (2023) are constructive and implementable.

**Fully integrated drift and support estimation**   The plug-in approach is statistically sound, but it is rather unclear if it will lead to convincing results in actual implementations. Indeed, part of the empirical success of standard diffusion models is explained by an implicit adaptation to the data support via the learned score networks Stanczuk et al. (2024). In this spirit, we propose a third strategy that only relies on information obtainable from the approximation of the log-gradient of the reverse $h$-transform. This is based on the intuition that $\nabla \log \overleftarrow{h}(x)$ must explode when $x$ approaches the low-dimensional data support $\Omega$ to ensure that the reverse diffusion process is killed in finite time. To formalise this intuition in the following, let us first give a useful characterisation of $\nabla \log \overleftarrow{h}$. The proof is given in Appendix C.

**Lemma 8.** *For all $x \notin \operatorname{supp} \alpha$, it holds that*

$$\nabla \log \overleftarrow{h}(x) = \mathbb{E}\left[\nabla_x \log G_r(x, Z_{\zeta-}^{\overleftarrow{h}}) \mid Z_0^{\overleftarrow{h}} = x\right]. \tag{4}$$

*In particular, for $\mathbb{P}_\alpha(Z^h_{\zeta_-} \in \cdot)$-a.e. $x$, it holds that*

$$\nabla \log \overleftarrow{h}(x) = \mathbb{E}_\alpha \left[ \nabla_x \log G_r(Z^h_0, x) \mid Z^h_{\zeta_-} = x \right]. \tag{5}$$

**Remark 9.** (i) If the forward process is simply an exponentially killed diffusion (i.e., $h = 1$), the representation of $\nabla \log \overleftarrow{h}$ as a conditional expectation can be used to motivate a simplified denoising score matching objective that only requires simulation of the forward process at its exponential lifetime, cf. (Christensen et al., 2025, Theorem 2.7) for the case of killed Brownian motion.

(i) Formula (5) is analogue to the well-known score representation

$$\nabla \log p_t(x) = \mathbb{E}_\alpha \left[ \nabla_x \log p_t(\vec{X}_0, x) \mid \vec{X}_t = x \right]$$

in conventional diffusion models for a homogeneous forward process $\vec{X}$ with transition densities $p_t(x, y)$, which is linked to Tweedie's formula Robbins (1956).

Let us for the moment consider the Green kernel $G_r$ of a Brownian motion. Then, we have

$$\nabla_x \log G_r(x, y) = -\sqrt{2r} \operatorname{sign}(x - y) \frac{K_{\nu+1}(\sqrt{2r}\|x - y\|)}{K_\nu(\sqrt{2r}\|x - y\|)}, \quad \nu = \frac{d - 2}{2}, \operatorname{sign}(x) := \frac{x}{\|x\|},$$

and therefore from above, for any $x \notin \operatorname{supp} \alpha$,

$$\nabla \log \overleftarrow{h}(x) = \sqrt{2r} \, \mathbb{E}^x \left[ \operatorname{sign}\left(Z^{\overleftarrow{h}}_{\zeta_-} - x\right) \frac{K_{\nu+1}(\sqrt{2r}\|Z^{\overleftarrow{h}}_{\zeta_-} - x\|)}{K_\nu(\sqrt{2r}\|Z^{\overleftarrow{h}}_{\zeta_-} - x\|)} \right] \approx \frac{d}{e} \, \mathbb{E}_x \left[ \frac{\operatorname{sign}(Z^{\overleftarrow{h}}_{\zeta_-} - x)}{\|Z^{\overleftarrow{h}}_{\zeta_-} - x\|} \right], \tag{6}$$

for large $d$, cf. 10.41 in DLMF. A more careful analysis furthermore reveals that the scaling factor $d$ in (6) can be replaced by $d - \dim \operatorname{supp} \alpha$ under mild regularity assumptions on the data manifold. This demonstrates quite clearly how the reverse process is dragged onto the data manifold. To exploit this connection for our purposes, we can use the explosive behavior of $\nabla \log \overleftarrow{h}$ near $\operatorname{supp} \alpha$ to obtain an insightful characterisation of the backward killing time, as the following result shows.

**Proposition 10.** *Suppose that*

*(i) the drift $b$ of $Z$ is locally bounded and its diffusion coefficient $\sigma\sigma^\top$ is uniformly elliptic;*

*(ii) $\operatorname{supp} \alpha$ is polar for both $Z^h$ and $Y$ solving $\sigma(Y_t) \, \mathrm{d}B_t$.*

*Then, the killing time $\zeta$ of $\overleftarrow{Z^h}$ is a.s. given by*

$$\zeta = \inf \left\{ t \geq 0 : \sup_{s \leq t} \left\| \nabla \log \overleftarrow{h}(\overleftarrow{Z^h_s}) \right\| = \infty \right\} = \inf \left\{ t \geq 0 : \left\| \nabla \log \overleftarrow{h}(\overleftarrow{Z^h}) \right\|_{L^2([0,t])} = \infty \right\}.$$

*Proof.* The proof proceeds along the same lines as the one of Theorem 2.8 in Christensen et al. (2025), where the result is proved for the particular case of the forward process $Z^h$ being given by an exponentially killed Brownian motion. $\square$

Most importantly in our context, the above considerations lead to simple and interpretable stopping criteria for the reverse process based on an estimate $s_{\widehat{\theta}}$ of $\nabla \log \overleftarrow{h}(x)$. For instance, when $Z$ is a Brownian motion and we target killing on the slightly enlarged data support $\Omega_\varepsilon$, then $Z^{\overleftarrow{h}}_{\zeta_-} \in \Omega$ implies for any $x \notin \Omega_\varepsilon$ that

$$\|\nabla \log \overleftarrow{h}(x)\| \approx \frac{d}{e} \left\| \mathbb{E}_x \left[ \frac{\operatorname{sign}(Z^{\overleftarrow{h}}_{\zeta_-} - x)}{\|Z^{\overleftarrow{h}}_{\zeta_-} - x\|} \right] \right\| \leq \frac{d}{e} \, \mathbb{E}_x \left[ \frac{1}{\|Z^{\overleftarrow{h}}_{\zeta_-} - x\|} \right] \leq \frac{d}{e\varepsilon}.$$

---

**Algorithm 4** Full generative algorithm without separate support estimation

---

**Input:** data $\{y_i\}_{i=1}^n \overset{iid}{\sim} \alpha$, $N \in \mathbb{N}$, $\varepsilon > 0, \underline{T} \geq 0, \gamma > 0, r > 0$, (approximation of) Green kernel $G_r$ of diffusion $Z$, enlarged data support $\Omega_\varepsilon$, forward transform $h$, backward initialization $\beta' \approx \mathbb{P}_\alpha(Z_{\zeta_-}^h \in \cdot)$, function class $\mathcal{S}$.

**for** $j = 1$ **to** $N$ **do**

    Draw $y_{i_j}$ uniformly with replacement from $\{y_i\}_{i=1}^n$ and simulate path $(z_t^{h,i_j})_{t\in[0,\zeta^{i_j})}$ of $h$-transform $Z^h$ started in $y_{i_j}$;

**end for**

**Training:** Learn $\widehat{\theta} = \arg\min_{\theta\in\mathcal{S}} \widehat{\mathcal{L}}(\theta)$ for

$$\widehat{\mathcal{L}}(\theta) = \frac{1}{N}\sum_{j=1}^N \int_{\underline{T}\wedge\zeta^{i_j}}^{\zeta^{i_j}} \left\| s_\theta(z_t^{h,i_j}) - \nabla_1 \log G_r(z_t^{h,i_j}, y_{i_j}) \right\|^2 dt$$

**Generation:**

Draw $x$ from $\beta'$

Simulate path $(y_t^{\widehat{\theta}})_{t\in[0,\widehat{\zeta_\varepsilon}]}$ of $Y^{\widehat{\theta}}$ initialised in $x$ until $\widehat{\zeta_\varepsilon} = \inf\{t \geq 0 : \|s_{\widehat{\theta}}(y_t^{\widehat{\theta}})\| \geq \gamma d\varepsilon^{-1}\}$

**Output:** $\widehat{\theta}, y_{\widehat{\zeta}}^{\widehat{\theta}}$

---

Combining this with Proposition 10, we may therefore reasonly stop the data-generating mechanism as soon as $\|s_{\widehat{\theta}}(Y_t^{\widehat{\theta}})\| \gtrsim \frac{d}{\varepsilon}$. This translates into a score-based reverse first killing time

$$\widehat{\zeta_\varepsilon} := \inf\left\{t \geq 0 : \|s_{\widehat{\theta}}(Y_t^{\widehat{\theta}})\| \geq \gamma d\varepsilon^{-1}\right\},$$

for some hyperparameter $\gamma$ or, equivalently, the first hitting time of the implicit data support estimator

$$\widehat{\Omega_\varepsilon} := \left\{x \in \mathbb{R}^d : \|s_{\widehat{\theta}}(x)\| \geq \gamma d\varepsilon^{-1}\right\}.$$

Such a stopping rule encodes mathematically, what a human observer would do intuitively when tracking the sequentially generated data:

> *stop the generation process as soon the observed ordered set of pixels visually resembles a high quality image.*

To obtain an estimator $s_{\widehat{\theta}}$ based on Proposition 7 without prior estimation of $\Omega_\varepsilon$, we simply replace the lower time index $\sigma_\varepsilon := \sup\{t < \zeta : Z_t^h \in \Omega_\varepsilon\}$ in the training objective with $\zeta \wedge \underline{T}$ for some hard-coded small value $\underline{T} \geq 0$. For specific forward processes such as exponentially killed diffusions, more elaborate alternatives are theoretically justified, see (Christensen et al., 2025, Theorem 2.7), Remark 9 and the experimental Section 6. The full generative algorithm is given in Algorithm 4.

**On model implications in absence of the polarity hypothesis** The polarity assumption is very flexible in the "true" lower intrinsic dimensional support setup since it also allows for complex situations such as unions of manifolds of varying intrinsic dimension. However, the strict manifold hypothesis is often considered to be an idealised mathematical simplification with a soft version that only assumes the data to be concentrated in thin tubular neighborhoods of manifolds being regarded as more appropriate for modern image datasets (Pope et al., 2021; Loaiza-Ganem et al., 2024). It is thus natural to ask, how our theory and algorithms can be adjusted if the support is thin but non-singular. Mathematically, this is a complex question and definite answers will strongly depend on the properties of the Lebesgue data density and the forward process in this scenario. By Lemma 8 and the explosive behaviour of the Green kernel on its diagonal, the magnitude of the score $\nabla \log \bar{h}(x)$ at a location $x$ can still be regarded as a good predictor for the proximity of $x$ to supp $\alpha$. The empirical stopping mechanism via tracking estimated score blow-up therefore remains

a good proxy in order to generate samples on or near the data support. However, the theoretical killing time of the backward process is no longer a simple first hitting time, since the forward model moves through the non-singular data support for a positive amount of time before termination. Even though this leads to less clean theoretical statements, it is important to observe that if the support is thin, the probability of a transient forward process, say exponentially killed Brownian motion in $d \geq 3$, to quickly leave the data support and then never return before killing is high. In this scenario, the first support entrance time and thus the first blow-up time of the backward process therefore remain good approximations of the true killing time and no adjustments for the suggested model implementation are needed.

### 4.5  Noise-Schedules as a special case of radial $h$-transforms

In classical diffusion models, the forward noise schedule $\alpha_t$ is of key importance for sampling. In our time-homogeneous setting this effect can be reproduced in high dimension by choosing a suitable $h$-transform for the forward process, exemplified here for a Brownian motion $Z$.

Consider the standard forward noising

$$x_t = \sqrt{\alpha_t}\, x_0 + \sqrt{1 - \alpha_t}\, z, \qquad z \sim \mathcal{N}(0, I_d),\ \|x_0\| = o(\sqrt{d}),$$

for which the law of large numbers gives $\|x_t\|/\sqrt{d} \longrightarrow \rho^*(t) := \sqrt{1 - \alpha_t}$ as $d \longrightarrow \infty$. The limiting radial trajectory $\rho^*(t)$ satisfies

$$\dot{\rho}^*(t) = -\frac{\dot{\alpha}_t}{2\sqrt{1 - \alpha_t}}. \tag{7}$$

To reproduce this within our framework, define $\Psi$ by

$$\Psi'(\rho) = \tfrac{1}{2\rho} - \dot{\rho}^*((\rho^*)^{-1}(\rho)), \qquad \Psi(\rho_0) = 0,$$

on a radial range corresponding to $t \in [\tau, T] \subset (0, \infty)$, where $\rho^*$ is smooth and strictly monotone. If we can choose a positive radial $r$-potential $h$ whose high-dimensional log-profile satisfies

$$-\frac{1}{d} \log h(x) \approx \Psi(\rho), \qquad \rho = \|x\|/\sqrt{d},$$

and assuming these log-asymptotics is differentiable on the considered radial range, then

$$\nabla \log h(x) \approx -\sqrt{d}\, \Psi'(\rho)\, \frac{x}{\|x\|}.$$

For $\rho_t = \|Z_t^h\|/\sqrt{d}$ we then have the approximation

$$d\rho_t \approx \Big( \frac{d-1}{2d\, \rho_t} - \Psi'(\rho_t) \Big)\, dt + \tfrac{1}{\sqrt{d}}\, dB_t$$

where $B$ is a one-dimensional Brownian motion. As $d \longrightarrow \infty$, we obtain the autonomous ODE

$$\dot{\rho} = \frac{1}{2\rho} - \Psi'(\rho) = \dot{\rho}^*((\rho^*)^{-1}(\rho)). \tag{8}$$

Hence, if we set $\rho(t) = \rho^*(t)$, then $(\rho^*)^{-1}(\rho(t)) = t$ (since $\rho^*$ is strictly monotone) and the autonomous ODE (8) reduces to $\dot{\rho}(t) = \dot{\rho}^*(t)$, i.e. (8) and (7) describe the same radial trajectory (possibly up to a time shift determined by the initial condition).

Thus the two descriptions differ in form: the classical schedule induces an explicit time-dependent drift for $\rho^*$, while the $h$-transform encodes the same effect through a space-dependent drift $-\Psi'(\rho)$. By the choice of $\Psi$, both ODEs (8) and (7) coincide along the trajectory $\rho^*(t)$ in the high-dimensional limit; consequently, in high dimensions, a single radial $h$ reproduces the schedule-driven radial behavior in a time-homogeneous formulation.

# 5 Features of the model

## 5.1 Natural conditioning

In the procedure described above, we have explained how to sample from the data distribution $\alpha$ using the time-homogeneous backward processes $Y = \overleftarrow{Z}^h$ or a learned version thereof when initialised from the terminal distribution of the forward process. The flexibility of our proposed model compared to existing diffusion models, however, arises from the observation that due to its time-homogeneous nature, the backward process $Y$ can be started in an interpretable way directly from *any* initial state $x$, regardless of the specific time, along the following lines:

1. Initialize the backward process $Y$ at a chosen state $x$, which could be:

   - A noisy version of a sample from $\alpha$, or
   - Any point of interest in the state space.

2. Simulate the backward process $Y$ until the lifetime $\zeta$, generating the trajectory $\{Y_t\}_{t \leq \zeta}$.

3. Extract the value $Y_{\zeta-}$ as the final sample. Note that this value lies within the support of $\alpha$, but is not necessarily distributed exactly according to $\alpha$.

4. Optionally, repeat the process for multiple initial states $x$ to analyze how the sampling distribution varies with the initialisation.

In Bayesian terminology, unconditional sampling corresponds to drawing from the prior distribution $\alpha$, whereas the current task involves sampling from $\mathbb{P}_x(Y_{\zeta-} \in dy)$, the posterior distribution conditioned on the input data $x$.

It is reasonable to expect that $Y_{\zeta-}$ will typically be close to $x$. Indeed, Proposition 5 states that

$$\mathbb{P}_x(Y_{\zeta-} \in dy) = \frac{1}{\overleftarrow{h}(x)} \frac{G_r(x,y)}{h(y)} \, \alpha(dy).$$

If $h(y)$ is approximately constant over the support of $\alpha$, the sampling procedure approximates drawing from the measure proportional to $G_r(x,y) \, \alpha(dy)$ (up to a normalizing constant). Typically, $G_r(x,y)$ decreases with the distance between $x$ and $y$, as illustrated in Remark 11. Consequently, points $y$ closer to $x$ appear with higher frequency than those farther away.

In other words, if the backward process $Y$ is started near the support of $\alpha$, the resulting sample from $\alpha$ will tend to be close to $x$. Here, the choice of the diffusion process $Z$ implicitly defines a corresponding notion of distance. So by choosing the starting point of $Y$, we get natural conditioning, and in exactly the same (learned) model as for unconditional sampling.

## 5.2 Fine tuning for conditional data generation

In addition to the natural conditioning properties discussed above, we may also use our model to explicitly sample from a conditional distribution of interest, which is only represented through a smaller subset of samples having certain properties. More specifically, suppose that instead of sampling from $\alpha$ we would like to sample from the posterior $\alpha(dy \mid u) \propto \pi(u \mid y) \, \alpha(dy)$, where $\pi(u)$ is the density of some latent random variable $U$ of interest and assume that the likelihood satisfies $\pi(u \mid \cdot) \in L^2(\alpha)$. To do so, we aim at lifting a generative model from our unconditional pre-trained model (i.e., using the learned score $\nabla \log \overleftarrow{h}$), for which more data is available. Ideally, we would like to simulate a backward process with drift determined by

$\nabla \log \bar{h}(\cdot \mid u)$, where $\bar{h}(x \mid u) := \int \frac{G_r(x,y)}{h(y)} \alpha(dy \mid u)$. This can be calculated similarly to Lemma 8 as follows

$$
\begin{aligned}
\nabla \log \bar{h}(x \mid u) &= \frac{\int \nabla_x G_r(x,y) \frac{1}{h(y)} \pi(u \mid y)\, \alpha(dy)}{\int G_r(x,y) \frac{1}{h(y)} \pi(u \mid y)\, \alpha(dy)} \\
&= \frac{\int \nabla_x \log G_r(x,y) \pi(u \mid y) \frac{G_r(x,y)}{\bar{h}(x)h(y)} \alpha(dy)}{\int \pi(u \mid y) \frac{G_r(x,y)}{\bar{h}(x)h(y)} \alpha(dy)} \\
&= \frac{\mathbb{E}_x\left[\nabla_x \log G_r(x, Z_{\zeta-}^{\bar{h}})\pi(u \mid Z_{\zeta-}^{\bar{h}})\right]}{\mathbb{E}_x\left[\pi(u \mid Z_{\zeta-}^{\bar{h}})\right]} \\
&= \mathbb{E}_x^u\left[\nabla_x \log G_r(x, Z_{\zeta-}^{\bar{h}})\right],
\end{aligned}
\tag{9}
$$

where $\mathbb{P}_x = \mathbb{P}(\cdot \mid Z_0^{\bar{h}} = x)$ and $\mathbb{P}_x^u$ is a probability measure defined by the Radon–Nikodym derivative

$$
\frac{d\,\mathbb{P}_x^u}{d\,\mathbb{P}_x} := \frac{\pi(u \mid Z_{\zeta-}^{\bar{h}})}{\mathbb{E}_x\left[\pi(u \mid Z_{\zeta-}^{\bar{h}})\right]},
$$

relative to $\mathbb{P}_x$. Recalling that Lemma 8 shows $\nabla \log h(x) = \mathbb{E}_x\left[\nabla_x \log G_r(x, Z_{\zeta-}^{\bar{h}})\right]$, we see that the conditional log gradient of the reverse $h$-transform is obtained via a change of measure induced by the likelihood $\pi(u \mid y)$ and we obtain the decomposition

$$
\nabla \log \bar{h}(x \mid u) = \nabla \log \bar{h}(x) + \mathbb{E}_x\left[\nabla_x \log G_r(x, Z_{\zeta-}^{\bar{h}}) \frac{\pi(u|Z_{\zeta-}^{\bar{h}}) - \mathbb{E}_x[\pi(u|Z_{\zeta-}^{\bar{h}})]}{\mathbb{E}_x[\pi(u|Z_{\zeta-}^{\bar{h}})]}\right].
$$

Furthermore, (9) yields

$$
\begin{aligned}
\nabla \log \bar{h}(x \mid u) &= \frac{\mathbb{E}_{Z_0^h \sim \alpha}\left[\nabla_x \log G_r(x, Z_0^h)\pi(u \mid Z_0^h) \mid Z_{\zeta-}^h = x\right]}{\mathbb{E}_{Z_0^h \sim \alpha}\left[\pi(u \mid Z_0^h) \mid Z_{\zeta-}^h = x\right]} \\
&= \int \nabla_x \log G_r(x,y)\, \mathbb{P}_{Z_0^h \sim \alpha(dy|u)}\left(Z_0^h \in dy \mid Z_{\zeta-}^h = x\right) \\
&= \mathbb{E}_{Z_0^h \sim \alpha(dy|u)}\left[\nabla_{Z_{\zeta-}^h} \log G_r(Z_{\zeta-}^h, Z_0^h) \mid Z_{\zeta-}^h = x\right],
\end{aligned}
$$

where we used that by Bayes' theorem,

$$
\pi(u \mid y)\, \mathbb{P}_{Z_0^h \sim \alpha}(Z_0^h \in dy \mid Z_{\zeta-}^h = x) \propto \mathbb{P}_{Z_0^h \sim \alpha(dy|u)}(Z_0^h \in dy \mid Z_{\zeta-}^h = x).
$$

This may be interpreted as an analogue of the score matching identity for the posterior score in diffusion models. Similarly to the fine-tuning approach from Denker et al. (2024), given a pre-processed estimator $s_{\hat{\theta}}$ of $\nabla \log \bar{h}$ and a smaller subset of training data for the conditional distribution $\alpha(dy \mid u)$, we may now aim at approximating only the difference

$$
C_u(x) := \nabla \log \bar{h}(x \mid u) - \nabla \log \bar{h}(x) = \mathbb{E}\left[\nabla_x \log G_r(x, Z_{\zeta-}^{\bar{h}}) \frac{\pi(u|Z_{\zeta-}^{\bar{h}}) - \mathbb{E}[\pi(u|Z_{\zeta-}^{\bar{h}})|Z_0^{\bar{h}}=x]}{\mathbb{E}[\pi(u|Z_{\zeta-}^{\bar{h}})|Z_0^{\bar{h}}=x]} \mid Z_0^{\bar{h}} = x\right],
\tag{10}
$$

of the log-gradient of the conditional and unconditional $h$-transform. From (Christensen et al., 2025, Lemma A.1), it follows that for any $\delta > 0$, $C_u$ coincides on $(\Omega_\delta)^c$ with the minimiser of

$$
C \mapsto \mathbb{E}_{Z_0^h \sim \alpha(dy|u)}\left[\left\|C(Z_{\zeta-}^h) + \nabla \log \bar{h}(Z_{\zeta-}^h) - \nabla_{Z_{\zeta-}^h} \log G_r(Z_0^h, Z_{\zeta-}^h)\right\|^2 \mathbf{1}_{(\delta,\infty)}(\|Z_{\zeta-}^h - Z_0^h\|)\right],
$$

in the space of measurable functions, where conditioning on $\{\|Z_{\zeta-}^h - Z_0^h\| > \delta\}$ ensures that the expectation is finite. Given $s_{\hat{\theta}} \approx \nabla \log \bar{h}$, this may be implemented as an empirical version of

$$
\underset{C \in \mathcal{C}}{\arg\min}\, \mathbb{E}_{Z_0^h \sim \alpha(dy|u)}\left[\left\|C(Z_{\zeta-}^h) + s_{\hat{\theta}}(Z_{\zeta-}^h) - \nabla_{Z_{\zeta-}^h} \log G_r(Z_0^h, Z_{\zeta-}^h)\right\|^2 \mathbf{1}_{(\delta,\infty)}(\|Z_{\zeta-}^h - Z_0^h\|)\right],
$$

---

**Algorithm 5** Conditional generation via fine tuning of unconditional algorithm

---

**Input:** data $\{y_i\}_{i=1}^n \overset{iid}{\sim} \alpha$, $M \in \mathbb{N}$, $\varepsilon, \delta, \gamma > 0, r > 0$, Green kernel $G_r$ of diffusion $Z$, forward transform $h$, backward initialisation $\beta_u' \approx \mathbb{P}_{\alpha(dy|u)}(Z_{\zeta_-}^h \in \cdot)$, function class $\mathcal{C}$, unconditional log-gradient $h$-transform estimator $\widehat{s} = s_{\widehat{\theta}} \approx \nabla \log \overleftarrow{h}$.

**for** $j = 1$ **to** $M$ **do**

Draw $y_{i_j}$ uniformly from $\{y_1, \ldots, y_n\}$ and simulate terminal value $z_{\zeta_-^{i_j}}^{h,i_j}$ of $h$-transform $Z^h$ started in

$y_{i_j} \sim \alpha(dy \mid u)$;

**end for**

**Training:** Learn $\widehat{C}_u \in \arg\min_{C \in \mathcal{C}} \widehat{\mathcal{L}}_u(C)$ for

$$\widehat{\mathcal{L}}_u(C) = \frac{1}{M} \sum_{j=1}^M \| C(z_{\zeta^{ij}_-}^{h,i_j}) + \widehat{s}(z_{\zeta_-^{i_j}}^{h,i_j}) - \nabla_{z_{\zeta_-}^{h,i_j}} \log G_r(z_{\zeta^{ij}_-}^{h,i_j}, y_{i_j}) \|^2 \mathbf{1}_{(\delta, \infty)}(\|z_{\zeta^{ij}_-}^{h,i_j} - y_{i_j}\|)$$

set $\widehat{s}(\cdot \mid u) = \widehat{C}_u + s_{\widehat{\theta}}$

**Generation:**

Draw $x$ from $\beta_u'$

Simulate path $(\widehat{y}_t^u)_{t \in [0, \widehat{\zeta}_\varepsilon]}$ of

$$dY_t^u = (b(Y_t^u) + \sigma\sigma^\top(Y_t^u)\widehat{s}(Y_t^u \mid u))\, dt + \sigma(Y_t^u)\, dW_t, \quad Y_0^u = x,$$

until $\widehat{\zeta}_\varepsilon = \inf\{t \geq 0 : \|\widehat{s}(\widehat{Y}_t^u \mid u) \geq \gamma d\varepsilon^{-1}\}$

**Output:** $\widehat{y}_{\widehat{\zeta}_\varepsilon}^u$

---

for some class of approximating functions $\mathcal{C}$. Given such minimiser $\widehat{C}_u$, we then define a conditional score estimator as

$$\widehat{s}(y \mid u) := \widehat{C}_u(y) + s_{\widehat{\theta}}(y).$$

The corresponding conditional generation procedure is summarised in Algorithm 5.

As an alternative to this fine-tuning approach, we may use the probabilistic characterisation (9) directly, provided that we have access to the likelihood $\pi(u \mid y)$ or a good approximation thereof (e.g., a classifier). To this end, we may first approximate the expected likelihood function

$$x \mapsto \mathbb{E}_x[\pi(u \mid Z_{\zeta_-}^{\overleftarrow{h}})] = \mathbb{E}_{Z_0^h \sim \alpha}[\pi(u \mid Z_0^h) \mid Z_{\zeta_-}^h = x]$$

which minimises the $L^2$-loss

$$\ell_u \mapsto \mathbb{E}_{Z_0^h \sim \alpha}\left[|\ell_u(Z_{\zeta_-}^h) - \pi(u \mid Z_0^h)|^2\right],$$

in $L^2(\alpha)$. Denote the solution of the associated nonparametric regression problem (based on a subset $\{y_1, \ldots, y_{n_1}\}$ of the unconditional data set $\{y_1, \ldots, y_n\}$ and some class of candidate likelihood functions $\mathcal{P}_u$) by $\widehat{\ell}_u$, and similarly, using the same reasoning as above based on (Christensen et al., 2025, Lemma A.1), estimate $\nabla \log \overleftarrow{h}(\cdot \mid u)$ by a minimiser of

$$s_u \mapsto \frac{1}{M} \sum_{j=1}^M \left\| s_u(z_{\zeta^{ij}_-}^{h,i_j}) - \nabla_2 G_r(y_{i_j}, z_{\zeta^{ij}_-}^{h,i_j}) \frac{\pi(u \mid y_{i_j})}{\widehat{\ell}_u(z_{\zeta^{ij}_-}^{h,i_j}) \vee \varepsilon} \right\|^2 \mathbf{1}_{\left\{\|z_{\zeta^{ij}_-}^{h,i_j} - y_{i_j}\| > \delta\right\}}$$

within some approximating function class $\mathcal{S}_u$. Here, $\delta, \varepsilon$ are regularisation parameters, the indices $i_j$ are uniformly drawn with replacement from the remaining data indices $\{n_1 + 1, \ldots, n\}$ and $z_{\zeta^{ij}_-}^{h,i_j}$ is a simulated value of the $h$-transform $Z^h$ at its lifetime when initialised in the data point $y_{i_j}$. Note here that the estimation procedure neither requires an estimator of $\nabla \log \overleftarrow{h}$ nor explicit access to samples of $\alpha(dy \mid u)$. The separate estimation of the denominator $\ell_u$ may however cause numerical challenges. These are more pronounced if

$\pi(u \mid Z_0^h)$ is concentrated around 0 under $\mathbb{P}_{Z_0^h \sim \alpha}$ corresponding to scarce data availability for the condition $U = u$.

### 5.3 Distances to the distribution and anomaly detection

We now address the question of how to measure the distance of an input $x$ from the data distribution $\alpha$. The previous discussion provides a natural measure for this, namely the time it takes the forward process $Y$ to transform the input $x$ into a sample from $\alpha$.

We can identify $x$ as an anomaly if the average lifetime exceeds a certain threshold $\underline{T}$ using the Monte-Carlo procedure given in Algorithm 6.

---

**Algorithm 6** Anomaly detection

---

**Input:** state $x$ to be evaluated, learned score $s_{\widehat{\theta}}$ based on Algorithm 4, threshold $\underline{T} > 0$, # of Monte-Carlo runs $N$, precision parameters $\gamma, \varepsilon > 0$
Initialize $\overline{\zeta} = 0$
**if** $\|s_{\widehat{\theta}}(x)\| < \gamma d\varepsilon^{-1}$ **then**
  **for** $i = 1$ **to** $N$ **do**
    Initialize $Y_0^{\widehat{\theta}} = x$
    **repeat**
      Simulate next step of $Y^{\widehat{\theta}}$
    **until** $\|s_{\widehat{\theta}}(Y_t^{\widehat{\theta}})\| \geq \gamma d\varepsilon^{-1}$
    Update $\overline{\zeta} \leftarrow \overline{\zeta} + t/N$
  **end for**
**end**
**if** $\overline{\zeta} > \underline{T}$ **then**
  Classify as anomaly
**end**

---

This can also serve as a criterion for determining whether new training data necessitates a modification of our trained model or not.

### 5.4 Class sampling and classification

The model can be naturally extended when the data can be decomposed into subclasses, i.e., when $\alpha$ is a mixture of distributions $\alpha_1, \ldots, \alpha_K$ with disjoint supports $\Omega^i$ corresponding to different classes $u_i$. In this scenario, there is a corresponding decomposition $\overleftarrow{h} = \sum_{i=1}^{K} \overleftarrow{h}_i$. The lifetime $\zeta$ is then determined as the minimum of the first entry times $\zeta_i$ into the supports of the $\alpha_i$. Combining unconditional sampling with class-conditional fine-tuned models from Section 5.2, this provides information about the specific class from which the sampled image originates and can thus be exploited for classification *without* pretrained classifier:

The conditional corrector $C_{u_i}(\overleftarrow{Z_t^h})$ in (10) represents the class-specific steering adjustment required to pull the state $\overleftarrow{Z_t^h}$ towards the class-$u_i$ sub-manifold $\Omega_i$. If the true underlying class of $\overleftarrow{Z_0^h}$ is $u_i$, the unconditional trajectory will naturally align with class $u_i$'s score direction, meaning the corrector magnitude $\|C_{u_i}(\overleftarrow{Z_t^h})\|$ will be small. Conversely, if $\overleftarrow{Z_t^h}$ is a noisy version of a sample $Z_0$ belonging to class $u_j \neq u_i$, the corrector for class $u_i$ must perform significant work (yielding a large norm) to steer the sample to its sub-manifold. For a stopping time $\widetilde{\zeta} \leq \zeta$ chosen large enough such that $(\|C_{u_i}(\overleftarrow{Z_t^h})\|)_{t \leq \widetilde{\zeta}}$ contains sufficient information on the implicit steering force along the unconditional path $(\overleftarrow{Z_t^h})_{t \in [0, \zeta)}$, we therefore define the corrector energy

$$E(u_i) := \int_0^{\widetilde{\zeta}} \|C_{u_i}(\overleftarrow{Z_t^h})\| \, \mathrm{d}t,$$

and let

$$u^* := \underset{u_i}{\arg\min}\, E(u_i)$$

be the minimal energy classifier of the class $u$ underlying the noisy initialisation $\overleftarrow{Z}_0^h$. The corresponding empirical classification algorithm is given in 7 and a concrete implementation is showcased in Section 6.5.

---

**Algorithm 7** Classification

---

**Input:** state $x$ to be classified, classes $i = 1, \ldots, K$, learned backward model $Y^{\widehat{\theta}}$ based on Algorithm 3, learned correctors $\widetilde{C}_{u_i}$ based on fine-tuning algorithm 5

Initialize $Y_0^{\widehat{\theta}} = x$

Simulate $(Y_t^{\widehat{\theta}})_{t \in [0, \widetilde{\zeta}]}$

**for** $i = 1$ **to** $K$ **do**

    calculate corrector energy $\widetilde{E}(u_i) = \int_0^{\widetilde{\zeta}} \|\widetilde{C}_{u_i}(Y_t^{\widehat{\theta}})\| \, \mathrm{d}t$

**end for**

**Output:** classification $u^* = \arg\min_{u_i} \widetilde{E}(u_i)$

---

The presented conditional diffusion model thus naturally enables classification through transfer learning. Refinements are also conceivable. For instance, by starting several runs from $x$ we can estimate the conditional class distribution, which allows the construction of statistical tests. Analogously to the case of anomaly detection, the average lifetime $\overline{\zeta}$ can then be regarded as a measure of reliability.

**On scale-invariant alternatives** A natural alternative to the energy $E(u)$ is to evaluate directional alignment, such as the integrated cosine similarity between the corrector and the unconditional score direction. While this approach is scale-invariant by construction, we find empirically that it yields inferior classification accuracy. This is primarily because directional metrics discard the magnitude of the corrector, which directly quantifies the steering effort required to reach a class-specific submanifold. Furthermore, in regions where the corrector magnitude is small (indicating high class compatibility), the corrector direction becomes highly sensitive to score estimation noise, leading to high-variance classification boundaries.

### 5.5 General transfer learning

The idea of transfer learning presented in the previous section can be applied more generally. For instance, if the classes are not predefined, but clusters are to be learned from the data, the typical approach is to perform clustering with known algorithms only on the support of $\alpha$, which is a significantly lower-dimensional task. Similar approaches may be possible for tasks in Reinforcement Learning.

## 6 Experiments

In this section we conduct experiments on the MNIST dataset (with its predefined training set of 60,000 samples) (LeCun, 1998) to highlight the practical relevance of ADDMs introduced in this paper. We emphasise that these experiments are a proof of concept. A full evaluation study with comparisons against baselines on more sophisticated image datasets is left to future work.

We focus on implementing the variance exploding forward model discussed in Section 4 by choosing $r = 1/\sigma^2$ for a noise scale $\sigma > 0$ ($\sigma = 7$ in our training setup), $Z = W$ for a Brownian motion $W$ and $h \equiv 1$, making $Z^h$ a Brownian motion killed at an independent exponential time $\tau \sim \mathrm{Exp}(1/\sigma^2)$. Setting $\nu = (d-2)/2$, the Green kernel, cf. Remark 11, is then well-approximated by

$$G_{1/\sigma^2}(x, y) \approx \frac{1}{\sqrt{2\pi\nu}} \Big(\frac{\nu}{\pi e}\Big)^\nu \frac{\|x - y\|^{-2\nu}}{\sigma^2},$$

provided $\|x - y\| = o(\nu)$ and thus for large $d$

$$\nabla \log \overleftarrow{h}(y) = \frac{\nabla_y \int G_{1/\sigma^2}(x, y)\, \alpha(dx)}{\int G_{1/\sigma^2}(x, y)\, \alpha(dx)} \approx (d - 2) \frac{\int \frac{x-y}{\|x-y\|^d}\, \alpha(dx)}{\int \|x - y\|^{2-d}\, \alpha(dx)}.$$

Under the polarity hypothesis, $\text{supp}\,\alpha$ is typically lower-dimensional. If we further assume that its diameter is finite and of order $\sqrt{d}$, this justifies for $y = o(d)$ the projection approximation

$$\nabla \log \overleftarrow{h}(y) \approx (d-2)\frac{y^* - y}{\|y^* - y\|^2}, \tag{11}$$

for $y^* \in \arg\min_{z \in \text{supp}\,\alpha}\|y - z\|$, that is $\nabla \overleftarrow{h}$ points in the direction of the metric projection of $y$ onto $\text{supp}\,\alpha$ with increasing force for larger $d$ and increasing proximity of $y$ to $\text{supp}\,\alpha$. We refer to Christensen et al. (2025) for further details and theoretical implications on projection estimation. The approximation (11) now motivates constructing an estimator for the guiding drift $s = \nabla \log \overleftarrow{h}$ of the backward process by learning the projection $y^*$, similarly to standard diffusion models that are designed to predict the noise levels at different scales. By the time-homogeneous nature of our model and the geometric interpretation of the score $s$, the noise level of a sample $y$ is characterised by its distance $\text{r} = \text{r}(y)$ to $\text{supp}\,\alpha$. We therefore design our estimators as

$$s_\theta(y, \text{r}) = (d-2)\frac{D_\theta(y, \text{r}) - y}{\text{r}^2 + \varepsilon},$$

where, following ideas from Karras et al. (2022), the U-Net projection estimator $D_\theta(y, \text{r})$ is modelled by

$$D_\theta(y, \text{r}) = c_{\text{skip}}y + c_{\text{out}}(\text{r})\text{UNet}(c_{\text{in}}(\text{r})y, \text{r}; \theta), \tag{12}$$

with coefficients

$$c_{\text{in}}(\text{r}) = \frac{1}{\sqrt{\sigma_{\text{data}}^2 + \text{r}_d^2}}, \quad c_{\text{skip}}(\text{r}) = \frac{\sigma_{\text{data}}^2}{\sigma_{\text{data}}^2 + \text{r}_d^2}, \quad c_{\text{out}} = \frac{\text{r}_d\sigma_{\text{data}}}{\sqrt{\sigma_{\text{data}}^2 + \text{r}_d^2}},$$

where $\text{r}_d = \text{r}/\sqrt{d}$ encodes the effective noise scale and $\sigma_{\text{data}}^2$ is a variance estimate of the clean data. As $\text{r}_d \to 0$ we have $c_{\text{skip}} \to 1$ and $c_{\text{out}} \to 0$. Close to the data support, where a well-learned model should have small effective noise scale $\text{r}_d$, the predictor thus increasingly concentrates around the input state $y$. The input scaling $c_{\text{in}}$ is chosen to approximately normalise the input variance. By the memoryless property of the exponential distribution, the score matching objective can be simplified to only require simulation of the terminal value $Z_\tau^h = Z_0^h + W_\tau$ of the forward model, where $W_\tau \sim \text{Laplace}(0, \sigma^2 I_d)$ (see Christensen et al. (2025)) and for the above parametrisation is given by

$$\mathcal{L}(\theta) = \mathbb{E}_{Z_0^h \sim \alpha}\left[\|s_\theta(Z_\tau^h, \text{r}(Z_0^h, Z_\tau^h)) - \nabla_{Z_\tau^h} \log G_{1/\sigma^2}(Z_0^h, Z_\tau^h)\|^2 \mathbf{1}_{(\delta, \infty)}(\text{r}(Z_0^h, Z_\tau^h))\right], \quad \text{r}(Z_0^h, Z_\tau^h) = \|Z_\tau^h - Z_0^h\|,$$

where $\delta$ is a small early stopping parameter that prevents blow-up of the denoising score loss. Similarly to denoising diffusion models, where a time-dependent and increasing weighting factor is introduced to achieve balanced learning across all noise scales, we also slightly adapt the denoising score loss for implementations by introducing a spatial weighting factor

$$\lambda(\text{r}(Z_0^h, Z_\tau^h)) = \frac{1}{\|\nabla_{Z_\tau^h} \log G_{1/\sigma^2}(\text{r}(Z_0^h, Z_\tau^h))\|^2 + \widetilde{\varepsilon}}\mathbf{1}_{(\delta, \infty)}(\text{r}(Z_0^h, Z_\tau^h)),$$

and set

$$\mathcal{L}_{\text{score}}(\theta) = \mathbb{E}_{Z_0^h \sim \alpha}\left[\lambda(\text{r}(Z_0^h, Z_\tau^h))\|s_\theta(Z_\tau^h, \text{r}(Z_0^h, Z_\tau^h)) - \nabla_{Z_\tau^h} \log G_{1/\sigma^2}(Z_0^h, Z_\tau^h)\|^2\right].$$

To approximate this score loss during training, we rely on importance sampling. This is necessary since $\|W_\tau\|$ scales with $\sigma\sqrt{d}$, which implies that in high dimensions a naïve Monte–Carlo estimator doesn't see sufficiently many samples at small noise scales close to $\delta$, rendering score estimation close to the data support difficult. This, however, is fundamentally important given that our adaptive termination criterion relies on accurately tracking the blow-up of the score $s$ near the manifold. We therefore don't sample the forward termination time $\tau$ directly, but instead first sample the proposal $T = \sigma^2\overline{\tau}U^k$ for $\overline{\tau} \sim \text{Exp}(1)$ independent of $U \sim \mathcal{U}((0,1))$ and sufficiently large $k$ ($= 12$ in our implementations). The density of $T$ is given by

$$q(t) = \frac{t^{1/k-1}}{k\sigma^{2/k}}\Gamma\left(1 - \frac{1}{k}, \frac{t}{\sigma^2}\right), \quad t > 0,$$

for the incomplete Gamma Function $\Gamma(a, b)$, which gives the importance weight

$$w(t) = k\sigma^{2(1-1/k)}\frac{t^{1-1/k}e^{-t/\sigma^2}}{\Gamma(1-\frac{1}{k}, \frac{t}{\sigma^2})} \propto t^{1-1/k},$$

where the approximation is appropriate for $t$ sufficiently small. Since conditional on $T = t$, $\mathrm{r}(Z_0^h, Z_0^h + W_T) = \|W_t\| \approx \sqrt{td}$ in high dimensions, the importance weight can therefore be conditionally approximated in terms of r by

$$w(t) \propto \mathrm{r}(Z_0^h, Z_0^h + W_t)^{2-2/k}.$$

this yields the following importance sampling approximation $\widetilde{L}_{\text{score}}$ of the weighted score loss $L_{\text{score}}$:

$$\widetilde{\mathcal{L}}_{\text{score}}(\theta) = \mathbb{E}_{z_0 \sim \alpha, \overline{\tau} \sim \text{Exp}(1), u \sim \mathcal{U}(0,1) v \sim \mathcal{N}(0, I_d)}\left[\mathrm{r}^{2-2/k}\lambda(\mathrm{r})\|s_\theta(z_{\sigma,\overline{\tau},u,v}, \mathrm{r}) - \nabla_2 \log G_{1/\sigma^2}(z_0, z_{\sigma,\overline{\tau},u,v})\|^2\right],$$

$$\text{where } z_{\sigma,\overline{\tau},u,v} = z_0 + \sigma\sqrt{\overline{\tau}}u^k v, \ \mathrm{r} = \mathrm{r}(z_0, z_{\sigma,\overline{\tau},u,v}).$$

We emphasize that this loss can be implemented very efficiently since we do not need to simulate entire trajectories of the forward model but for every drawn data sample only need to add noise once. Compared to standard denoising diffusion models there is therefore no additional computational cost for the forward pass simulation during training.

We then obtain a score estimator $\widehat{s} = s_{\widehat{\theta}}$ and corresponding projection estimator $\widehat{D} = D_{\widehat{\theta}}$ by optimising sample versions of

$$\mathcal{L}_{\text{total}}(\theta) = \widetilde{\mathcal{L}}_{\text{score}}(\theta) + \gamma \mathcal{L}_{\text{pen}}(\theta)$$

via an iterative Adam optimiser Kingma & Ba (2015) over 50 epochs, where $\mathcal{L}_{\text{pen}}(\theta)$ is a soft thresholding penalty calculated via the predictions $D_\theta(z, r(z_0, z))$ for simulated corruptions $z$ of a clean datapoint $z_0$ as described above. This penalisation is introduced to constrain the backward process to produce outputs in the admissible MNIST data range.

Given such estimators, we simulate the backward process via an Euler approximation of the SDE with r̄-adaptive step size selection. The latter is necessary to stabilise the algorithm since the learned drift blows up as the generated data approaches the true data support and constant step sizes thus increase the likelihood of overshooting the target domain. The distance input parameter r to the score estimator $\widehat{s}(y, \mathrm{r})$ is simply initialised in the Euclidean distance $\bar{\mathrm{r}}_0 = \|\bar{z}_0\|$ of the intialisation $z_0$, which is motivated by the fact that MNIST pictures are dominated by black pixels and thus have small Euclidean norm. We iteratively update $\bar{\mathrm{r}}_t$ by setting $\bar{\mathrm{r}}_{t_{i+1}} = \|\bar{z}_{t_{i+1}} - \widehat{D}(\bar{z}_{t_i}, \bar{\mathrm{r}}_{t_i})\| \vee \delta$, preserving time-homogeneity of the generative process. Here we clamp at $\delta$ since noised training data with distances smaller than $\delta$ are discarded in training. In our implementation we choose $\delta = 0.28$, which on MNIST corresponds to an effective output noise scale of $\delta/\sqrt{d} = 0.28/\sqrt{784} = 0.01$, that is, we target generation for slightly noisy images with noise standard deviation $\sigma_{\min} = 0.01$, which is a standard early stopping parameter in denoising diffusion models. In accordance with our theoretical findings, we terminate the algorithm at time $t_i$ if either $\|\widehat{s}(\bar{z}_{t_i})\|$ (or, for numerical stability, a rolling average over the last $m$ samples for small $m \in \mathbb{N}$) surpasses a large threshold $M$, or, if $\bar{\mathrm{r}}_{t_i} \leq 1.05\delta$. The latter is a numerical safeguard needed because the clamping of $\bar{\mathrm{r}}$ at $\delta$ can lead to the explosion criterion not being triggered if we initialise at low noise levels where the denoiser will become gradually closer to identity by design, which can lead to $\|\widehat{D}(\bar{z}_{t_i}, \bar{\mathrm{r}}_{t_i})\| \leq \delta^2$, thence making $1/\bar{\mathrm{r}}_{t_i}$ a more robust predictor of the score norm. At termination we output $\widehat{D}(\bar{z}_{t_i}, \bar{\mathrm{r}}_{t_i})$. Let us point out, however, that in our experiments we find no significant performance improvements compared to outputting the terminal SDE value $\bar{z}_{t_i}$ directly, showing that termination indeed happens close to the true data support. Motivated by (11) we set $M = \beta\frac{d-2}{\delta}$ for a tuning parameter $\beta > 0$. If $\bar{\mathrm{r}}_{t_i}$ decreases below a predetermined threshold $\mathrm{r}_{\text{ode}}$, we further stabilise the generative algorithm by deactivating Gaussian noise injection and instead let the process follow the probability flow ODE associated to the drift $\widehat{s}$. In our experiments we set this threshold value to 1.5.

## 6.1 Unconditional Generation

We run our unconditional generative model with $\beta = 0.70$ and initialise with high (relative to MNIST data variance) Gaussian noise $\mathcal{N}(0, 4^2 I_d)$ . While initialising the algorithm with high variance Laplace noise as

suggested by the theoretical time-reversion framework also works, empirically a Gaussian initialisation yields slightly better results at lower computational costs due to faster convergence of the algorithm. Given the adaptive nature of the generative framework it would be interesting to systematically optimise the noise initialisation in future work. Examples of generated digits are provided in Figure 1.

Unconditional generation with Gaussian initialisation

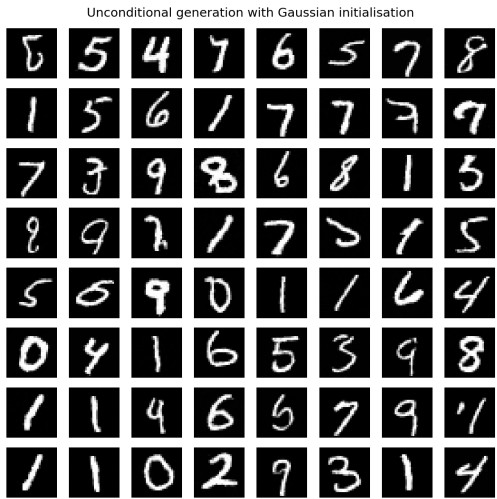

Figure 1: Examples of generated digits

Based on $10,000$ generated samples we report in Table 1 LeNet-FID scores (Heusel et al., 2017), classifier confidence (both based on a pretrained LeNet-5 classifier LeCun et al. (1998)) and mode entropy of our model and compare them against DDPM (Ho et al., 2020) and DDIM (Song et al., 2021a) generators. Both our model and the DDPM implentation represent lightweight architectures of the same order of magnitude ($2.5 \times 10^6$ parameters for our model compared to $6.1 \times 10^6$ parameters for the DDPM). Optimisation is performed with the Adam optimiser for the same number of epochs (50) to enable a fair comparison. On average our adaptively terminated model stopped after $\sim 146$ steps with a standard deviation of 4.6 steps, where we discarded runs that hadn't stopped after 1,000 steps (the observed 86.21% success rate can be improved at a slight performance loss by decreasing $\beta$). We use the DDPM model as a high-budget comparison with 1,000 SDE solver steps and adapt the DDIM without retraining to the learned DDPM model with 150 ODE steps to match the average NFE budget of our model.

| Model | NFE ↓ | LeNet-FID ↓ (min = 0) | Classifier Confidence ↑ (max = 100.00%) | Mode entropy ↑ (max ≈ 2.30) |
|---|---|---|---|---|
| **High-Budget** | | | | |
| DDPM baseline | 1,000 | **2.5803** | 94.96% | **2.2994** |
| **Matched-Budget** | | | | |
| DDIM | 150 | 24.7387 | 91.51% | 2.2597 |
| **ADDM (ours)** | $145.9 \pm 4.6$ | **9.1379** | **95.69%** | **2.2731** |

Table 1: Unconditional generative performance of our model compared to high-budget DDPM baseline and matched-budget DDIM baseline

In this setup we find that our model strongly outperforms the matched-budget DDIM model, while the DDPM high-budget baseline yields a higher FID score and comparable mode entropy, but has a slightly worse classifier confidence. These results demonstrate that our adaptive generative algorithm can produce high-quality, easily recognisable digit samples at comparatively low computational cost and based on more compact architectures with 60% fewer parameters than the DDPM/DDIM.

## 6.2 Adaptive denoising

In all our denoising experiments we set the termination parameter $\beta = 1.2$ and use the same model trained for unconditional generation. The generation was initialised in given noisy images and terminated according to the same score explosion criterion used for unconditional generation. Figure 2 depicts simulation results for different noise corruptions (Gaussian, Laplace, Uniform and Salt & Pepper impulse noise) at noise scales $\sigma \in \{0.6, 0.9\}$.

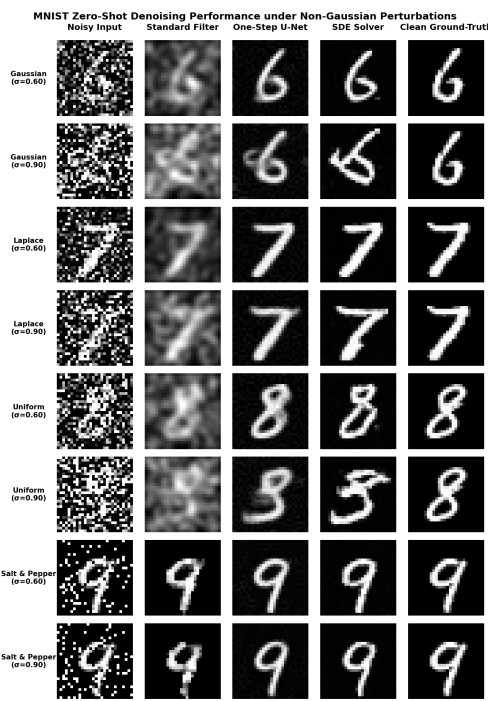

Figure 2: Denoising results for different noise corruption types and noise scales

Remarkably, even though the denoiser model $D_\theta(y, \mathrm{r})$ is trained strictly on Laplace noise distribution, the time-homogeneous SDE solver generalises zero-shot to entirely different noise distributions at test time. We report Peak-Signal-to-Noise-Ratios (PSNR) and $L^2$ reconstruction errors in Table 2 (averaged over 1,000 test samples each), with performance metrics of a standard Gaussian filter included for scale. The fact that our one-step denoiser consistently outperforms our SDE solver although the latter produces crisper outputs is a typical manifestation of the perception-distortion trade-off (Blau & Michaeli, 2018).

Our model's robustness to different noise corruptions stands in sharp contrast to standard denoising diffusion models, which are fragile to non-Gaussian noise since at test time the physical noise level of the observed data needs to be related to the remaining time steps of the solver while the trained model is optimised for Gaussian noise schedules.

Crucially, our noise scale parameter $\bar{\mathrm{r}}_{t_i}$ is not locked to a deterministic clock but is dynamically re-evaluated at each step based on the actual physical distance between the current state and the predicted clean projection of the previous state. Because the solver uses this physical distance feedback loop, the SDE acts as a robust stochastic projection operator. Even when the test-time noise is non-Laplace, the solver simply calculates the predicted distance to the manifold at each step, contracts the state stochastically, and stops when the trajectory has successfully landed close to the clean data manifold (triggered by the score norm explosion). This dynamic self-correction renders the solver fairly immune to both initial noise estimation errors and out-of-distribution noise distributions.

| Noise Type | Model / Method | PSNR (dB) ↑ | $L^2$ Reconstruction Error ↓ |
|---|---|---|---|
| Gaussian | Noisy Input | 8.25 | 10.83 |
| | Gaussian Filter | 11.74 | 7.25 |
| | **Our One-Step Model** | **19.27** | **3.10** |
| | **Our ADDM Solver** | 16.90 | 4.08 |
| Laplace | Noisy Input | 9.11 | 9.82 |
| | Gaussian Filter | 12.61 | 6.56 |
| | **Our One-Step Model** | **20.40** | **2.72** |
| | **Our ADDM Solver** | 18.31 | 3.48 |
| Uniform | Noisy Input | 7.49 | 11.82 |
| | Gaussian Filter | 10.98 | 7.92 |
| | **Our One-Step Model** | **18.06** | **3.57** |
| | **Our ADDM model** | 15.51 | 4.81 |
| Salt & Pepper | Noisy Input | 10.18 | 8.69 |
| | Gaussian Filter | 18.20 | 3.50 |
| | **Our One-Step Model** | **21.89** | **2.30** |
| | **Our ADDM model** | 19.83 | 2.92 |

Table 2: Denoising performance for different noise types ($\sigma = 0.60$, $N = 1,000$ samples)

The model's zero-shot capacity to denoise out-of-distribution data is further explored Figure 3. We see that the model is capable of upscaling pixelated images and denoise elastically warped digits without forcing the output to match seen training data.

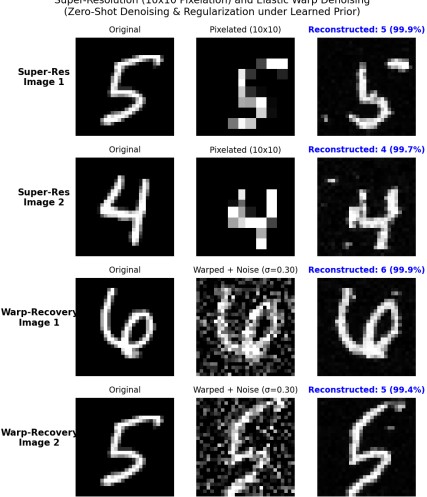

Figure 3: Super-resolution of pixelated images and denoising of affinely warped MNIST digits. Reconstruction percentages are based on a pretrained LeNet-5 classifier

In Figure 4 we track the first two principal components of a denoising SDE path for noised digits 0 and 1 and compare them against linear interpolations between the ground truth and their noisy versions. We see that denoising is highly efficient with paths exhibiting little curvature and closely tracking the idealised linear interpolation that crosses through non-blurry digit regions in PCA space.

### 6.3 On-the-fly fine-tuning for inpainting and class-conditional steering

We first test the fine-tuning approach from 5.2 in an image inpainting experiment, see Figure 5.

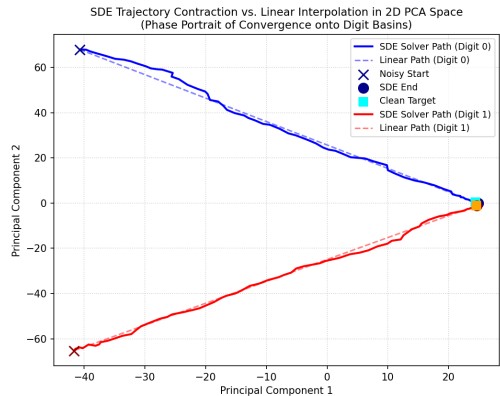

Figure 4: Trajectories of SDE generator in 2D PCA space for digits 0 and 1

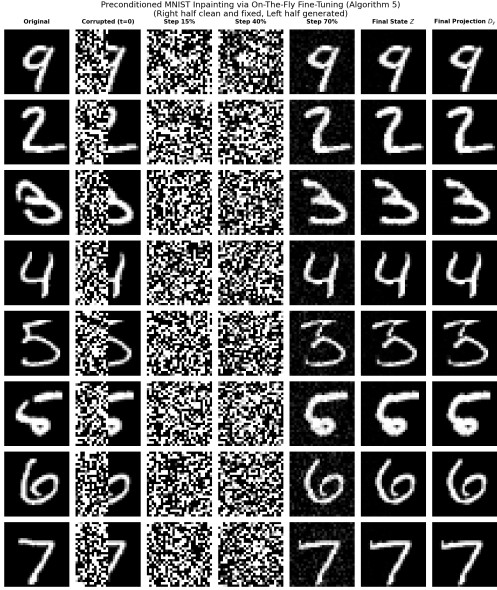

Figure 5: Inpainting of masked digits via on-the-fly fine tuning

Here we are given different images $z_0$, where the left half is pure Gaussian noise ($\sigma = 4.0$; equivalent to a black mask of the left half, which we noise at test time) and the right half is a clean digit. For every $z_0$ we search the training set of 60,000 clean digits for the top $K = 1,000$ nearest neighbors based on the $L^2$ distance of their clean right halves. This yields a local training subset representing the conditional distribution $\alpha(dx \mid u)$ for a right half class conditioning $u$.

We then parameterise the conditional score as:

$$s_\vartheta(y, \mathrm{r} \mid u) \coloneqq s_{\mathrm{uncond}}(y, \mathrm{r}) + \frac{U_{u,\vartheta}(y, \mathrm{r})}{\mathrm{r}^2 + \varepsilon} \tag{13}$$

where $U_{u,\vartheta}$ is a preconditioned U-Net constructed as in (12) and $s_{\mathrm{uncond}}$ is our pretrained model for unconditional generation. We then train the model on the minibatch of $K$ nearest neighbours according to Algorithm 5, where as for unconditional generation we perform importance sampling and introduce the same spatial score loss weighting to enhance numerical performance. This on-the-fly training for different inputs $z_0$ is computationally efficient because of the small sample size $K$ with an average training time of 15 seconds on a standard Apple M4 integrated GPU. To keep the U-Net inputs in-distribution during SDE updates,

the clean right half is perturbed with isotropic noise at the current noise scale:

$$\breve{z}_{t,\text{right}} \leftarrow z_{0,\text{right}} + \frac{\breve{r}_t}{\sqrt{d}}v_t, \quad v_t \sim \mathcal{N}(0, I_{d/2}). \tag{14}$$

The noisy left half is updated according to the reverse SDE with learned conditional score $\widehat{s}_u$. Since conditionally on the initialisation, coordinates are independent in the forward process, this is a mathematically rigorous representation of the conditional reverse trajectory. The SDE simulations terminate at a similar rate as unconditional generation with less than 190 steps and a 100% termination rate. As shown in Figure 5 the algorithm yields excellent results, completing all but one of the half-digits to clean versions of their original.

As a second example for conditional generation, we test whether starting from a noisy 8 (Laplace corruptions with $\sigma = 0.9$) we can run a fine-tuned conditional generative model that outputs a 3 instead. To this end, on the training subset of class 3 digits we train the fine-tuned drift (13) according to Algorithm 5 and for the optimised parameter $\widehat{\vartheta}$ set the drift to

$$s_{\widehat{\vartheta}}(y, r \mid u) \coloneqq s_{\text{uncond}}(y, r) + \eta \frac{U_{u,\widehat{\vartheta}}(y, r)}{r^2 + \varepsilon}.$$

Here, upscaling the guidance scale $\eta$ (Dhariwal & Nichol, 2021; Ho & Salimans, 2022) allows to steer the generative process stronger towards the targeted subclass of 3 digits. Figure 6 displays the result for guidance scales of $\eta = 0$ (unconditional denoising) and $\eta \in \{17, 25\}$ (moderate to strong steering). As observed before, the unconditional denoising algorithm recovers clean images of the ground truth. For $\eta = 17$, 50% of the digits are successfully transformed into 3s, with the success rate increasing to 70% in the stronger steering regime $\eta = 25$ albeit at a slight visual degradation in image quality.

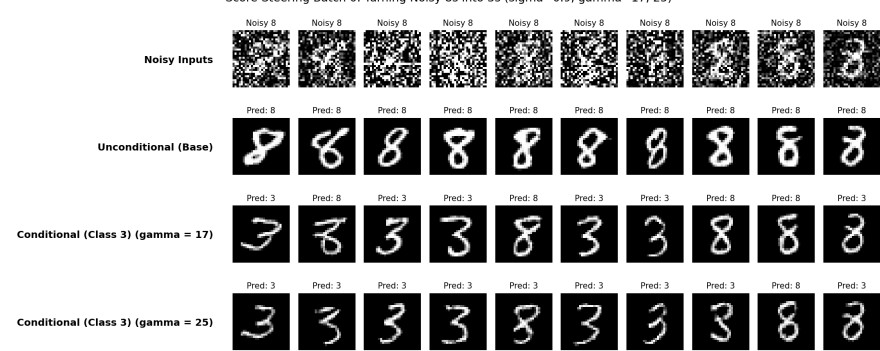

Figure 6: Steering noisy 8s into 3s via conditional fine-tuning

## 6.4 Anomaly detection

We test the model's ability to classify a given sample as an anomaly based on the observed simulation time. Since the tested samples will have similar initial noise levels as the clean MNIST data to make the task harder, we decide to terminate the algorithm as soon as the value $(d-2)/\|\widehat{D}(\breve{z}_{t_i}, \breve{r}_{t_i}) - \breve{z}_{t_i}\|$, which tracks score blow-up more sensitively at lower noise levels, surpasses the threshold $M$. Since our specific implementation of the score estimator also yields estimated projection denoisers $\widehat{D}(y, r)$, we can also base anomaly tests on the initial projection distance $\|\widehat{D}(\breve{z}_0, \breve{r}_0) - \breve{z}_0\|$ of a given sample $\breve{z}_0$ as a simple simulation-free statistic. In Table 3 we report AUC (Area Under Curve) statistics of both methods based on comparing 1,000 samples (each) of clean MNIST against clean FashionMNIST (OOD) (Xiao et al., 2017) and noisy MNIST data, where we choose a light Gaussian noise scale $\sigma = 0.1$ to make the detection task harder.

Both metrics exhibit large AUC scores close to 1 demonstrating their suitability for discriminating between clean and corrupted or OOD data, where the projection distance is not only computationally more efficient

| Category | solver steps (mean) | AUC (steps) | $\|\widehat{D}(\breve{z}_0, \breve{r}_0) - \breve{z}_0\|$ (mean) | AUC (projection dist.) |
|---|---|---|---|---|
| MNIST clean | $1.4 \pm 5.4$ | – | $2.3965 \pm 0.4366$ | – |
| MNIST noisy ($\sigma = 0.1$) | $20.2 \pm 1.6$ | 0.9474 | $3.9670 \pm 0.2043$ | **0.9994** |
| FashionMNIST | $22.0 \pm 6.0$ | 0.9640 | $5.8168 \pm 1.6088$ | **0.9793** |

Table 3: AUC statistics for simulation steps and projection distance based anomaly detection

but also more accurate. Interestingly, the results show that the algorithm terminates quite quickly when run on FashionMNIST samples as well and the initial projection distance is not too large. This indicates that the denoiser leverages shared higher-order geometric structures and spatial priors rather than performing simple pixel-level matching. This is further visualised in Figure 7

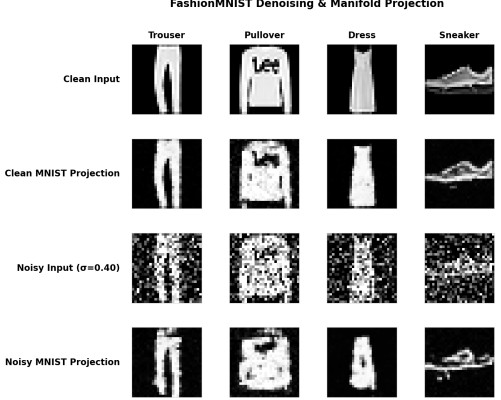

Figure 7: Running the ADDM (learned on MNIST) on clean and noisy FashionMNIST samples without retraining

## 6.5 Classification

As a final experiment, we use our unconditional generative model together with the learned guidance correctors $\widehat{C}_u(y, r) = \frac{U_{u,\vartheta}(y,r)}{r^2 + \varepsilon}$ of fine-tuned models for classes $u$, cf. Sections 5.2 and 6.3, to implement the energy corrector classification Algorithm 7.

For computational classification efficiency, it is sufficient to not fully denoise the sample but only drive it close enough to the clean digit mode. We therefore simulate the unconditional model only until $\widetilde{\zeta} := \inf\{t_j : \breve{r}_{t_j} \le r_{\min}\}$, where we set $r_{\min} = 3.0 \gg 0.28 = \delta$ in our runs. For every considered class $u$ we then calculate the empirical corrector energy

$$\widehat{E}(u) := \sum_{t_j \le \widetilde{\zeta}} \|\widehat{C}_u(\breve{z}_{t_j}, \breve{r}_{t_j})\| \Delta t_j \mathbf{1}_{[r_{\min}, r_{\max}]}(\breve{r}_{t_j}),$$

where restricting to $\breve{r}_{t_j} \le r_{\max}$ ($r_{\max} = 15$ in our runs) is introduced because of potential inaccurate learning at high noise levels. To control for structural bias caused by possibly differing unconditional score estimation accuracies in separate digit mode environments, we simulate a rough approximation $\widetilde{B}(u)$ of

$$\widehat{B}(u) = \mathbb{E}_{z_0 \sim \alpha(\cdot|u), r \sim \mathcal{U}([r_{\min}, r_{\max}]), \varepsilon \sim \mathcal{N}(0, r^2 I_d/d)} \left[ \|\widehat{C}_u(z_0 + \varepsilon, r)\| \right]$$

based on a minibatch of 100 class-$u$ samples. The debiased corrector energy is then calculated as

$$\widetilde{E}(u) := \frac{\widehat{E}(u)}{\widehat{B}(u)} = \sum_{t_j \le \widetilde{\zeta}} \|\widetilde{C}_u(\breve{z}_{t_j}, \breve{r}_{t_j})\| \Delta t_j \mathbf{1}_{[r_{\min}, r_{\max}]}(\breve{r}_{t_j}), \quad \text{where } \widetilde{C}_u(\breve{z}_{t_j}, \breve{r}_{t_j}) := \frac{\widehat{C}_u(\breve{z}_{t_j}, \breve{r}_{t_j})}{\widehat{B}(u)}$$

and the final classification is defined by

$$u^* = \arg\min_u \widetilde{E}(u).$$

Specifically, we now consider digit classes $u \in \{3, 8\}$ and use Algorithm 7 with the above empirical statistics to binary classify 1,000 corrupted digits with Laplace noise at noise level $\sigma \in \{0.6, 0.9\}$. We compare the results against two classification methods that rely on a pre-trained LeNet-5 classifier: (i) direct LeNet-5 classification of the noisy input $\tilde{z}_0$ and (ii) first denoise with our unconditional generative model and then classify the denoised output with LeNet-5. Results are summarised in Table 4.

| Method | Accuracy | Total evaluation time |
|---|---|---|
| **Noise scale $\sigma = 0.60$** | | |
| LeNet-5 Classifier | 80.00% | **0.04s** |
| Denoise-then-Classify | **94.10%** | 74.23s |
| **Corrector Energy** | 87.30% | 65.54s |
| **Noise scale $\sigma = 0.90$** | | |
| LeNet-5 Classifier | 65.40% | **0.04s** |
| Denoise-then-Classify | 82.90% | 78.32s |
| **Corrector Energy** | **83.10%** | 68.05s |

Table 4: Binary classification performance given 1,000 corrupted 3 and 8 digits under moderate and heavy Laplace noise

As expected from our denoising experiments in Section 6.2, Denoise-then-Classify yields very high accuracy for the moderate noise level $\sigma = 0.6$ since the model works exceptionally well as a denoiser in this regime. Our corrector energy method performs worse, but at higher speed since no full denoising is performed, and strongly outperforms the one-step LeNet-5 classifier, which is cheapest to evaluate. However, our classifier-free corrector energy method turns out to be more robust to the noise level, having the highest accuracy with a comparatively much smaller relative drop in performance when upscaling the noise level to $\sigma = 0.9$. Under such heavy noise, Denoise-then-Classify suffers from stochastic semantic morphing, where noisy 3s are occasionally denoised into realistic 8s and vice versa. The corrector energy method sidesteps this problem by evaluating class compatibility along the trajectory before the semantic morphing is locked in.

## 7 Conclusion

We have introduced a novel class of adaptive denoising diffusion models grounded in Doob's $h$-transform. This theoretical framework provides a general and adaptable foundation for time-homogeneous diffusion models that eliminates the necessity to model artificial time dependencies and thereby enhances interpretability. A key innovation in our model is the introduction of the polarity hypothesis, which closely parallels the manifold hypothesis in machine learning. This property enables an intuitive and efficient mechanism for determining the termination of the denoising process, addressing a crucial challenge in implementing our generative model.

Another notable feature of our framework is a methodological simplification for learning the dynamics of the denoising process. Unlike conventional diffusion models that require estimating temporally inhomogeneous dynamics, we must learn a time-*independent* backward drift, which is achieved by a denoising score matching procedure. While this comes at a cost of having to learn the now random simulation time for the generative process as well, we argue that no separate estimation strategy is needed for this purpose, but that a simple explosion criterion for the estimated backward drift along the generated path yields an adaptive termination rule.

The time-homogeneous nature of the model opens up numerous opportunities for practical applications, which we demonstrated with several experiments on MNIST. While its utility will depend on the specific requirements of each use case, the theoretical framework established here provides a robust foundation for further experimental exploration, particularly in the context of transfer learning.

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

## A Doob's $h$-transform and lifetime of a diffusion process

The following is based on Chung & Walsh (2005), which provides a level of generality necessary for our purposes. For a more explicit discussion in the univariate case, see also Borodin & Salminen (2002).

We first note that the Green kernel can be found explicitly for the Brownian motion:

**Remark 11.** Let $Z = W$ be a standard $d$-dimensional Brownian motion, $m$ Lebesgue measure, then

$$p_t(x, y) = (2\pi t)^{-d/2} \exp\left(-\frac{|x-y|^2}{2t}\right).$$

In this case, (Erdélyi et al., 1954, p. 146) yields

$$G_r(x, y) = G_r(|x-y|) = (2\pi)^{-d/2} 2 \left(\frac{|x-y|^2}{2r}\right)^{(2-d)/4} K_{(d-2)/2}\left(|x-y|\sqrt{2r}\right),$$

where $K_\nu$ is the modified Bessel function of the second kind as defined in Lebedev (1972), p. 109. The function $G_r(\cdot)$ has a pole in 0 and is decreasing.

Now we give the remaining proofs:

*Proof of Proposition 3.* The finiteness of $\zeta$ follows from (Chung & Walsh, 2005, Theorem 13.50):

$$\mathbb{P}_x(\zeta < \infty) = \int_0^\infty \int e^{-rt} \frac{1}{h(x)} p_t(x, y)\kappa(dy)dt = \frac{1}{h(x)} \int G_r(x, y)\kappa(dy) = 1.$$

The second statement follows by using the generator identity

$$\mathbb{A}^h f(x) = \frac{1}{h(x)} \left(\mathbb{A}(h(x)f(x)) - rh(x)f(x)\right),$$

taking into account that $h$ is $r$-harmonic (i.e., $(\mathbb{A} - r)h(x) = 0$) outside the support of $\beta$.

The third statement follows from (Chung & Walsh, 2005, Theorem 13.39) (after correction of an obvious typo), where the special case $x = x_0$ yields

$$\mathbb{P}_{x_0}(Z_{\zeta-}^h \in dy) = \frac{G_r(x_0, y)}{h(x_0)}\kappa(dy) = G_r(x_0, y)\kappa(dy) = \beta(dy).$$

$\square$

*Proof of Proposition 5.* The first claim holds by combining (Chung & Walsh, 2005, Theorem 13.34) with Proposition 3. The other two are direct consequences of Proposition 3. $\square$

## B Stochastic control

For the proof of Proposition 4, we start with the following

**Lemma 12.** *Let $h$ be as above and $g(x) := -\log h(x)$. Then, outside of the support of $\beta$, it holds that*

$$\mathbb{A}g(x) + r - \frac{1}{2}\nabla g(x)^\top \sigma(x)\sigma^\top(x)\nabla g(x) = 0.$$

*Proof.* Outside of $\mathrm{supp}\,\kappa$, it holds that $\mathbb{A}h(x) - rh(x) = 0$, and a straightforward calculation shows that

$$\mathbb{A}g(x) = -r + \frac{1}{2}\nabla g(x)^\top \sigma(x)\sigma^\top(x)\nabla g(x).$$

$\square$

Now, we follow the standard verification approach to stochastic stopping and control problems, see Øksendal & Sulem (2019), Chapter 5, where we assume the appropriate standard regularity assumptions for the objects without explicitly mentioning them. Outside of $\operatorname{supp}\kappa$, using the notation

$$\mathbb{A}^u v(x) = \langle \sigma(x)u + b(x), \nabla v(x)\rangle + \frac{1}{2}\operatorname{Tr}\left[\Sigma(x)\nabla^2 v(x)\right],$$

where $\Sigma(x) = \sigma(x)\sigma(x)^\top$, it holds

$$\inf_u \left(\mathbb{A}^u v(x) + k(u)\right) = \langle b(x), \nabla v(x)\rangle + \frac{1}{2}\operatorname{Tr}\left[\Sigma(x)\nabla^2 v(x)\right] + r + \inf_u \left(\langle \sigma(x)u, \nabla v(x)\rangle + \frac{1}{2}\|u\|^2\right)$$

$$= \langle b(x), \nabla v(x)\rangle + \frac{1}{2}\operatorname{Tr}\left[\Sigma(x)\nabla^2 v(x)\right] + r - \frac{1}{2}\|\sigma(x)^\top \nabla v(x)\|^2$$

with minimiser $u = -\sigma(x)^\top \nabla v(x)$. Using $v(x) = g(x) = -\log h(x)$ and Lemma 12, we see that $g(x)$ fulfills the HJB equation for problem (1). Following the steps in the verification procedure in Øksendal & Sulem (2019), p. 92ff., it is easily seen that this solution to the HJB equation is indeed a solution to (1), proving the first part of Proposition 4.

Now, we study the connection to the KL divergence and write $Y = Z^h$. Note that

$$\left.\frac{d\mathbb{P}^{Z^u}}{d\mathbb{P}^Y}\right|_{\mathcal{F}_\zeta} = \exp\left(\frac{1}{2}\int_0^\zeta \|u_t\|^2 dt - M_\zeta\right),$$

for some (local) martingale $M$ with $M_0 = 0$. For the killed process $Y$, by the definition of the $h$-transform,

$$\left.\frac{d\mathbb{P}^Z}{d\mathbb{P}^Y}\right|_{\mathcal{F}_\zeta} = \frac{\mathrm{e}^{r\zeta}h(Z_0)}{h(Z_\zeta)}.$$

This yields

$$\left.\log\frac{d\mathbb{P}^{Z^u}}{d\mathbb{P}^Y}\right|_{\mathcal{F}_\zeta} = \left.\log\frac{d\mathbb{P}^{Z^u}}{d\mathbb{P}^Z}\frac{d\mathbb{P}^Z}{d\mathbb{P}^Y}\right|_{\mathcal{F}_\zeta} = \frac{1}{2}\int_0^\zeta \|u_t\|^2 dt + r\zeta + \log h(Z_0^u) - \log h(Z_\zeta^u) + M_\zeta.$$

We obtain that

$$\mathrm{KL}(\mathbb{P}_x^{Z^u}\big|_{\mathcal{F}_\zeta} \| \mathbb{P}_x^Y\big|_{\mathcal{F}_\zeta}) = \mathbb{E}_x \left.\log\frac{d\mathbb{P}^{Z^u}}{d\mathbb{P}^Z}\right|_{\mathcal{F}_\zeta} = J(u,x) + \log h(x) \geq v(x) + \log h(x) = 0$$

with equality for $u = u^*$, so that the KL divergence is the variational gap in problem (1).

## C  Score matching and manifold learning

*Proof of Proposition 7.* It holds that

$$\mathcal{L}_{\mathrm{ex}}(\theta) = \mathbb{E}\left[\int_0^{\zeta_\varepsilon} \|\nabla_y \log \overleftarrow{h}(\overleftarrow{Z_t^h}) - s_\theta(\overleftarrow{Z_t^h})\|^2 \, dt\right]$$

$$= \mathbb{E}\left[\int_0^{\zeta_\varepsilon} |s_\theta(\overleftarrow{Z_t^h})|^2 \, dt\right] - 2\mathbb{E}\left[\int_0^{\zeta_\varepsilon} \langle \nabla_y \log \overleftarrow{h}(\overleftarrow{Z_t^h}), s_\theta(\overleftarrow{Z_t^h})\rangle \, dt\right] + C, \tag{15}$$

where $C$ is independent of $\theta$. Let $\widetilde{\beta}(dy) = \mathbb{P}_\alpha(Z_{\zeta-}^h \in dy)$. Since the time reversal of the $h$-transform $Z^{\overleftarrow{h}}$ started in $\widetilde{\beta}$ is equal in law to the $h$-transform $Z^h$ started in $\alpha$, we obtain from Proposition 5 that

$$\frac{G_r(x,y)}{\overleftarrow{h}(x)}\widetilde{\beta}(dx) = h(y)\frac{G_r(y,x)}{h(y)\overleftarrow{h}(x)}\widetilde{\beta}(dx) = h(y)\,\mathbb{P}_y(Z_{\zeta-}^h \in dx).$$

Using this and the assumed integrability conditions, we can calculate the second term as follows:

$$\mathbb{E}\Big[\int_0^\zeta \langle \nabla_y \log \bar{h}(\bar{Z}_t^h), s_\theta(\bar{Z}_t^h)\rangle \, dt\Big] = \int\int \langle \nabla_y \log \bar{h}(y), s_\theta(y)\rangle \frac{\bar{h}(y)}{\bar{h}(x)} G_r(x,y)\, m(dy)\, \widetilde{\beta}(dx)$$

$$= \int \langle \nabla_y \log \bar{h}(y), s_\theta(y)\rangle \bar{h}(y) h(y)\, m(dy)$$

$$= \int \langle \nabla_y \bar{h}(y), s_\theta(y)\rangle h(y)\, m(dy)$$

$$= \int\int \langle \nabla_y \frac{G_r(y,z)}{h(z)}, s_\theta(y)\rangle h(y)\, m(dy)\, \alpha(dz)$$

$$= \int\int \langle \nabla_y \log G_r(y,z), s_\theta(y)\rangle \frac{h(y)}{h(z)} G_r(y,z)\, m(dy)\, \alpha(dz)$$

$$= \int\int \langle \nabla_y \log G_r(y,z), s_\theta(y)\rangle \frac{h(y)}{h(z)} G_r(z,y)\, m(dy)\, \alpha(dz)$$

$$= \int \mathbb{E}_z\Big[\int_0^\zeta \langle \nabla_{Z_t^h} \log G_r(Z_t^h, Z_0^h), s_\theta(Z_t^h)\rangle \, dt\Big]\, \alpha(dz)$$

$$= \mathbb{E}_\alpha\Big[\int_0^\zeta \langle \nabla_{Z_t^h} \log G_r(Z_t^h, Z_0^h), s_\theta(Z_t^h)\rangle \, dt\Big].$$

This gives

$$\mathbb{E}\Big[\int_0^{\zeta_\varepsilon} \langle \nabla_y \log \bar{h}(\bar{Z}_t^h), s_\theta(\bar{Z}_t^h)\rangle \, dt\Big]$$

$$= \mathbb{E}\Big[\int_0^\zeta \langle \nabla_y \log \bar{h}(\bar{Z}_t^h), s_\theta(\bar{Z}_t^h)\rangle \, dt\Big] - \mathbb{E}\Big[\int_{\zeta_\varepsilon}^\zeta \langle \nabla_y \log \bar{h}(\bar{Z}_t^h), s_\theta(\bar{Z}_t^h)\rangle \, dt\Big]$$

$$= \mathbb{E}\Big[\int_0^\zeta \langle \nabla_{Z_t^h} \log G_r(Z_t^h, Z_0^h), s_\theta(Z_t^h)\rangle \, dt\Big] - \mathbb{E}\Big[\int_0^{\sigma_\varepsilon} \langle \nabla_y \log \bar{h}(Z_t^h), s_\theta(Z_t^h)\rangle \, dt\Big]$$

$$= \mathbb{E}\Big[\int_{\sigma_\varepsilon}^\zeta \langle \nabla_{Z_t^h} \log G_r(Z_t^h, Z_0^h), s_\theta(Z_t^h)\rangle \, dt\Big] + \mathbb{E}\Big[\int_0^{\sigma_\varepsilon} \langle \nabla_{Z_t^h} \log \tfrac{G_r(Z_t^h, Z_0^h)}{\bar{h}(Z_t^h)}, s_\theta(Z_t^h)\rangle \, dt\Big].$$

Moreover, the first term satisfies

$$\mathbb{E}\Big[\int_0^{\zeta_\varepsilon} \|s_\theta(\bar{Z}_t^h)\|^2 \, dt\Big] = \mathbb{E}\Big[\int_{\sigma_\varepsilon}^\zeta \|s_\theta(Z_t^h)\|^2 \, dt\Big].$$

Substituting these results into (15), we obtain

$$\mathcal{L}_{\mathrm{ex}}(\theta) = \mathbb{E}\Big[\int_{\sigma_\varepsilon}^\zeta \|\nabla_{Z_t^h} \log G_r(Z_t^h, Z_0^h) - s_\theta(Z_t^h)\|^2 \, dt\Big] - 2\mathbb{E}\Big[\int_0^{\sigma_\varepsilon} \langle \nabla_{Z_t^h} \log \tfrac{G_r(Z_t^h, Z_0^h)}{\bar{h}(Z_t^h)}, s_\theta(Z_t^h)\rangle \, dt\Big] + C',$$

where $C'$ is a constant independent of $\theta$. □

*Proof of Lemma 8.* Using (2) and Proposition 5, we find for any $x \notin \mathrm{supp}\,\alpha$,

$$\nabla \log \bar{h}(x) = \frac{1}{\bar{h}(x)} \int \nabla_x G_r(x,y) \frac{1}{h(y)}\, \alpha(dy) = \int \nabla_x \log G_r(x,y) \frac{G_r(x,y)}{\bar{h}(x)h(y)}\, \alpha(dy)$$

$$= \mathbb{E}_x\big[\nabla_x \log G_r(x, Z_{\zeta-}^{\bar{h}})\big].$$

By Proposition 5, the time reversed process $\overleftarrow{Z}^h$ has the same distribution as the diffusion process $Z^{\bar{h}}$ started in the forward terminal distribution $\beta_h = \mathbb{P}_\alpha(Z^h_{\zeta_-} \in \cdot)$. For any Borel sets $A, B$ it follows

$$\int_B \mathbb{P}_\alpha(Z^h_0 \in A \mid Z^h_{\zeta_-} = x) \, \mathbb{P}_\alpha(Z^h_{\zeta_-} \in dx) = \mathbb{P}_\alpha(Z^h_0 \in A, Z^h_{\zeta_-} \in B)$$

$$= \mathbb{P}_{Z^{\bar{h}}_0 \sim \beta_h}(Z^{\bar{h}}_{\zeta_-} \in A, Z^{\bar{h}}_0 \in B)$$

$$= \int_B \mathbb{P}(Z^{\bar{h}}_{\zeta_-} \in A \mid Z^{\bar{h}}_0 = x) \, \beta_h(dx)$$

$$= \int_B \mathbb{P}(Z^{\bar{h}}_{\zeta_-} \in A \mid Z^{\bar{h}}_0 = x) \, \mathbb{P}_\alpha(Z^h_{\zeta_-} \in dx),$$

and thus, $\mathbb{P}(Z^{\bar{h}}_{\zeta_-} \in dy \mid Z^{\bar{h}}_0 = x)$ is a regular conditional probability for $Z^h_0$ given $Z^h_{\zeta_-}$ w.r.t. $\mathbb{P}_\alpha$. Consequently, uniqueness of regular conditional probabilities gives

$$\mathbb{P}_\alpha(Z^h_0 \in dy \mid Z^h_{\zeta_-} = x) = \mathbb{P}(Z^{\bar{h}}_{\zeta_-} \in dy \mid Z^{\bar{h}}_0 = x), \quad \text{for } \mathbb{P}_\alpha(Z^h_{\zeta_-} \in \cdot)\text{-a.e. } x.$$

This together with (4) and symmetry of $G_r$ yields (5). $\qquad\square$

