# OpenReview forum: "Beyond Fixed Horizons: A Theoretical Framework for Adaptive Denoising Diffusions"
_TMLR — Under review for TMLR_

### Review · Reviewer_HFJy · 2026-05-25

**Summary Of Contributions:**

This paper proposes a new class of generative diffusion models that replaces the fixed, deterministic time horizon used in standard denoising diffusion models with a random horizon, obtained via Doob’s h-transform with respect to exponential killing times. A key consequence is that both the forward noising process and the backward denoising process become time-homogeneous: the learned drift $s_\theta(x)$ depends only on the current state $x$, rather than on an explicit time index $t$. This removes the need to learn temporally inhomogeneous dynamics and, in principle, allows the number of denoising steps to adapt to the noise level of each individual sample.

The work appears theoretically sound and offers an interesting, somewhat unconventional extension of current diffusion-model research.

The most obvious weakness is the complete absence of any empirical study, which makes it difficult to assess whether the proposed framework is practically viable. Beyond this, there are deeper structural concerns (regarding the robustness of the proposed stopping criterion, the feasibility of stable neural network training without signal normalization across scales, and the cost of simulation-based training), which I will elaborate on below. The writing is also highly technical and provides limited intuitive explanation, which may make it difficult for a broader machine learning audience to appreciate the paper's main contributions.

**Audience:**

Yes

**Audience Explanation:**

While practitioners may find the results in this paper difficult to understand or apply, readers with a strong theoretical background in related areas may find the ideas interesting. The work may also inspire further studies aimed at developing more practical methods.

**Broader Impact Concerns:**

I do not have concerns.

**Claims And Evidence:**

No

**Claims Explanation:**

While I do not see obvious errors in the theoretical claims, the paper provides no empirical evidence to support its claims beyond the theory. This makes it difficult to assess the applicability of the proposed method in practical settings. In particular:

1) Regarding the stop criterion in Proposition 8, the authors claim that the true score would explore near the data manifold. But this argument is only true for the ground truth score function instead of the network approximation. Actually, the past empirical study showed that the learnt score is least reliable near the data manifold. And in fact, no neural network can reliably represent it (and the training object in Algorithm 2 explicitly excludes this region). Additionally, there is no analysis to theoretically or empirically explain how the approximation errors in the estimated score propagate into the quality of the stopping time.

2) The paper basically suggests that the score function can be approximated by the neural network through minimizing the objective in Algorithm 4. However, past experience in existing diffusion model training suggests that a successful training heavily relies on a careful normalization of the signal strength.  The paper has no discussion on this issue or proposes any practical mitigation, leaving the practical trainability of the method an open question.

3) While algorithms 5 to 7 showcase some interesting applications of this method, there's no empirical evidence to support their effectiveness at all, not even on synthetic data.  In addition, Algorithm 7 actually requires the estimation of class supports, but it's unclear to me how we can access them in practice.

**Requested Changes:**

1. Empirical studies are needed to demonstrate the effectiveness of the proposed training methods. Please also provide empirical evidence supporting the effectiveness of the stopping criteria.

2. Please provide empirical studies that support the validity of the proposed downstream applications.

3. Please add a discussion of the computational cost of the forward-pass simulation. It would be helpful to give practitioners a clear understanding of the additional cost required by this component.

4. Ideally, please also consider adding more intuitive explanations of proposed theoretical results.

---

> ### Author Response · Authors · 2026-07-09
> **Rebuttal**
>
> We thank the reviewer for their valuable feedback and critical remarks that have led to a substantial revision of the manuscript and added experimental material. Below we address all points raised and requested changes in detail. Further comments summarising updates made in the revision are given above in our general answer to all reviewers.
>
> > Regarding the stop criterion in Proposition 8, the authors claim ...
>
> We thank the reviewer for these observations. While it is true that score estimation in standard diffusion models is more challenging at small times, early stopping procedures are introduced in practice to mitigate this issue, and several  studies (e.g. Oko et. al (2023), Tang and Yang (2024)) have shown that such thresholding allows for statistically optimal score estimation. Excluding a small tubular neighborhood $\Omega_\delta$ in the score estimation loss serves the same purpose in our model and therefore our algorithms as well as implementations only target generating data close to the data support $\Omega$. The score therefore only needs to be learned well outside of $\Omega_\delta$. To promote accurate learning in an environment of $\Omega_\delta$, which, as the reviewer points out, is crucial for learning the killing time well, we now introduce an importance sampling procedure in our implementations, which, however, comes at no additional computational costs in training; see also the general answer to the reviewers above. We strongly agree that a quantification of the score estimation loss propagation into the hitting time estimation error is a highly intriguing question, but, due to its mathematical complexity, is unfortunately out of scope of this paper.
>
> > The paper basically suggests that the score function can be approximated by the neural network through minimizing the objective...
>
> Thanks for raising this critical point. In our experiments, we found this to be a crucial aspect for ensuring stable learning across all noise scales and thus for optimising the model output as well. In the new experimental section, we therefore now describe in detail  how a pure spatial weighting  based on the log-gradient of the Green kernel can serve this purpose in our model and also point out in the theoretical section that such loss objective adjustments can be necessary in practice.
>
> >While algorithms 5 to 7 showcase some interesting applications of this method, there's no empirical evidence...
>
> Thanks for this critical remark and especially for pointing out that the suggested classification algorithm was not clearly embedded into the general modelling pipeline by requiring access to class supports estimates. We have therefore fully revised the classification algorithm and now introduce a method that is based on tracking the (learned) score correction $C_u$ for fine-tuned conditional sampling along an unconditional path of the generative algorithm started in a noisy sample to be classified. The rationale behind this is that the corrector encodes the steering adjustment needed to push the backward process from its unconditional path toward the class-$u$ submanifold, and thus needs to grow substantially if $u$ does not coincide with the true class underlying the data sample. The class $u^\ast$ minimizing the corrector energies can thus serve as class predictor based on a simulated unconditional  path. Full details along with experimental evidence (also for the general fine-tuning algorithm 5) are now provided in the paper, see also our general answer to all reviewers above.
>
> >Empirical studies are needed to demonstrate the effectiveness of the proposed training methods...
>
> We have now added extensive  experiments on MNIST in a new Section 6 that demonstrate the effectiveness of the model for unconditional generation and all downstream tasks described in theory. Section 6 also provides details on how to implement the model successfully.
>
> >Please provide empirical studies that support the validity of the proposed downstream applications.
>
> Empirical studies are now included. Please refer to the previous point and our general answer to the reviewers.
>
> >Please add a discussion of the computational cost of the forward-pass simulation...
>
> We thank the reviewer for this important suggestion. By considering an exponentially killed Brownian motion as a forward model, we show that the model training can be implemented at no additional costs compared to forward-pass simulations of standard diffusion models, since it is not necessary to simulate the full forward trajectory but only sample the forward process at its lifetime. This has been specifically emphasised in the experimental section now.
>
> >Ideally, please also consider adding more intuitive explanations...
>
> Thanks again for this suggestion. In the revised version, we have added further comments on theoretical results that had not been yet sufficiently explained.

---

### Review · Reviewer_8sNS · 2026-06-06

**Summary Of Contributions:**

The paper proposes a theoretical framework for diffusion models with random rather than fixed time horizons. Using Doob’s h-transform, the authors construct a time-homogeneous forward noising process and derive a corresponding time-homogeneous reverse denoising process whose stopping time can adapt to the sample. The paper claims that this is especially useful when the data lies on a low-dimensional or polar support, then the authors connects the method to the manifold hypothesis. The authors demonstrate their method with applications in unconditional generation, natural conditioning, conditional fine-tuning, anomaly detection, and classification.

**Audience:**

Yes

**Audience Explanation:**

This paper has developed a theory for adaptive denoising diffusion models, which is interseting for TMLR audience.

**Claims And Evidence:**

No

**Claims Explanation:**

This paper proposes intereseting theory. However, Ifeel like the broader claims about practical generative modeling, adaptive denoising, conditioning, anomaly detection, and classification are not yet convincingly supported. The paper does not include any empirical results, even on simple synthetic manifold data, so it is unclear whether the proposed training objective and stopping rule work in practice.

The theory also relies on strong assumptions, especially the polarity hypothesis and reliable estimation of a score that explodes near the data support. It is not clear whtether this will hold for finite, noisy, high-dimensional real datasets without empirical evidence.

**Requested Changes:**

I am not an expert in this field, here are my suggestions:
- It is strongly advised that the authors ddd experiments and compare with fixed-horizen diffusion baselines, at least on synthetic low-dimensional manifolds.
- State the assumptions of the main results more explicitly, including the integrability, regularity, polarity, and support conditions.
- Discuss what happens when data are noisy or only approximately low-dimensional, since this is the realistic setting for most machine learning datasets.

---

> ### Author Response · Authors · 2026-07-09
> **Rebuttal**
>
> We wish to express our gratitude to the reviewer for the time and effort that went into their review and their helpful and critical remarks. Specific requested changes are addressed below and a general answer to all reviewers is provided above that summarises our updated in the revised version:
>
> >It is strongly advised that the authors ddd experiments and compare with fixed-horizen diffusion baselines, at least on synthetic low-dimensional manifolds.
>
> We thank the reviewer for this suggestion. We have now added several experiments on MNIST to provide an empirical proof of concept for the theoretical claims we have made. Comparisons with fixed-horizon DDPM and DDIM baselines are provided for unconditional generation. This is summarised in a new Section 6, along with a detailed explanation how the model is implemented concretely. Please refer to the general answers provided to all reviewers above and the revised version for details.
>
> > State the assumptions of the main results more explicitly, including the integrability, regularity, polarity, and support conditions.
>
> In Section 3.1, we have now clearly stated regularity and symmetry conditions on the reference process $Z$ as separate assumptions. Proposition 7 now states the specific intregrability assumptions that are required. Polarity conditions on the support are part of every theoretical statement that needs them.
>
> > Discuss what happens when data are noisy or only approximately low-dimensional, since this is the realistic setting for most machine learning datasets.
>
> Thanks for this important remark. In Section 4.4, we have now included a detailed discussion on this point, which we copy here for the convenience of the reviewer:  The polarity assumption is very flexible in the ``true'' lower intrinsic dimensional support setup since it also allows for complex situations such as unions of manifolds of varying intrinsic dimension. However, the strict manifold hypothesis is often considered to be an idealised mathematical simplification with a soft version that only assumes the data to be concentrated in thin tubular neighborhoods of manifolds being regarded as more appropriate for modern image datasets (Pope et al., 2021, Loaiza-Ganem et al., 2024). It is thus natural to ask, how our theory and algorithms can be adjusted if the support is thin but non-singular. Mathematically, this is a complex question and definite answers will strongly depend on the properties of the Lebesgue data density and the forward process in this scenario. By Lemma 8 and the explosive behaviour of the Green kernel on its diagonal, the magnitude of the score $\nabla \log \overleftarrow{h}(x)$ at a location $x$ can still be regarded as a good predictor for the proximity of $x$ to $\operatorname{supp} \alpha$. The empirical stopping mechanism via tracking estimated score blow-up therefore remains a good proxy in order to generate samples on or near the data support. However, the theoretical killing time of the backward process is no longer a simple first hitting time, since the forward model moves through the non-singular data support for a positive amount of time before termination. Even though this leads to less clean theoretical statements, it is important to observe that if the support is thin, the probability of a transient forward process, say exponentially killed Brownian motion in $d \geq 3$, to quickly leave the data support and then never return before killing is high. In this scenario, the first support entrance time and thus the first blow-up time of the backward process therefore remain good approximations of the true killing time and  no adjustments for the suggested model implementation are needed.

---

### Review · Reviewer_EyWX · 2026-07-02

**Summary Of Contributions:**

This paper proposes a theoretical framework for diffusion models with flexible time horizons in both the noising and denoising processes, using Doob’s h-transform. The authors develop the stochastic-process foundations of the framework, including h-transforms, time reversal from exponential or random lifetimes, and an optimal-control interpretation based on KL divergence. The paper further discusses several potential downstream applications, including natural conditioning, conditional fine-tuning, anomaly detection, classification, and transfer learning.

**Audience:**

Yes

**Audience Explanation:**

The theory of diffusion generative models is an active topic within the TMLR community. This paper is likely to interest researchers working on the theoretical foundations of diffusion models. Additionally, the proposed framework suggests a promising direction for generative modeling with potential downstream applications, so researchers working on practical diffusion models may also find the paper relevant.

**Broader Impact Concerns:**

No concerns on the ethical implications of the work.

**Claims And Evidence:**

No

**Claims Explanation:**

My answer is between Yes and No. As a theoretical paper, the mathematical development is substantial and coherent. The paper clearly motivates why replacing fixed deterministic horizons with random lifetimes can lead to time-homogeneous reverse dynamics, and it provides several propositions connecting Doob’s h-transform, time reversal, stochastic control, KL divergence, and score-matching-style learning objectives. These results support the paper’s core theoretical claim that such a framework can be constructed.

However, diffusion models are widely studied largely because of their practical impact, so demonstrating theoretical properties alone is not fully sufficient. Empirical experiments supporting the proposed downstream applications would significantly strengthen the contribution and increase the impact of the paper.

**Requested Changes:**

The only request I have is empirical experiments, even a toy experiment would significantly strengthen the paper. For example,
(a) teh author could test the framework over known low-diemsnional manifolds, such as high dimension ball, low-rank Gaussian, and visualize the adaptive stopping behavior.
(b) a direct comparison with a fixed-horizon diffusion baseline. Using metrics to demonstrate the strength of the proposed framework.
(c) The authors could start the reverse process from a partially corrupted or partially observed sample and show that the final generated sample remains close to the initialization while moving toward the data distribution, using toy dataset like MNIST or Fashion-MNIST.

---

> ### Author Response · Authors · 2026-07-09
> **Rebuttal**
>
> We thank the reviewer for the time and effort that went into their review and their excellent suggestions that motivated us to strengthen the paper with convincing empirical evidence. Below we address the proposed changes. Further comments summarising updates made in the revision are given above in our general answer to all reviewers.
>
> In the newly added Section 6, we now give a detailed description how a particular variance exploding version of our general modelling pipeline can be efficiently implemented based on optimising a weighted denoising score loss on U-Net architectures. We compare the unconditional generative performance of the model with DDPM (high budget) and DDIM (matched step budget given that our model terminates after only 150 steps on average) and report FID, classifier confidence, and mode entropy scores.
>
> A denoising experiment, as suggested by the reviewer, has  been implemented with different noise corruptions to test the model's robustness and its zero-shot denoising capacity on unseen data. Furthermore, an inpainting experiment that showcases the conditional fine-tuning experiment (masked data as suggested by the reviewer) has been implemented as well.
>
> All other downstream tasks that we described are now also supported with experiments and evaluations. Further details are summarised in the general answer to all reviewers above, and the full picture is given in the uploaded revised version.

---

### Author Response · Authors · 2026-07-09
**Rebuttal: General response to all reviewers**

We thank the reviewers for their valuable and constructive feedback, which motivated us to extend the manuscript considerably. Since there has been a broad consensus among the reviewers that the paper would be considerably strengthened by supporting our theoretical claims with convincing empirical evidence, we have added an entire new section (Section 6) that

1. explains how a variance exploding formulation of our model (Brownian motion killed at an exponential time as forward model) can be efficiently implemented;
2. summarises experiments that we conducted for unconditional generation and all the downstream tasks (adaptive denoising, conditional generation via fine-tuning, anomaly detection and classification) described in Section 5 of the paper.

Specifically, we motivate a projection-based score estimator from first approximation principles in high dimensions. This allows to focus the neural-network training on learning a denoiser $D_\theta(y,\mathrm{r})$ whose input parameters are given by the state $y$ of the generative process and the distance $\mathrm{r}$ to the data support, which during training is encoded as the distance between a clean data sample and its noisy version. During generation, the distance $\mathrm{r}$ is updated state-adaptively by considering the distance between the current state and the projection of the previous state via the learned denoiser $D_\theta$. This achieves a time-homogeneous generative process as imposed by our theory, which we terminate according to our derived score explosion criterion, where the termination threshold is again motivated by first principles and the early stopping parameter $\delta$.

As pointed out by one of the reviewers, a careful normalisation of the signal strength in the denosing score loss is necessary for successful implementation of diffusion models. We found the same phenomenon to be true in our experiments. We therefore introduce a spatial weighting with the squared norm of the Green kernel's log-gradient to achieve stable training across all noise scales, which in our setting correspond to the normalised physical distances $\mathrm{r}/\sqrt{d}$ between clean and noisy data samples. Additionally, instead of using a na\"ive Monte-Carlo estimator of the weighted denoising score loss, we introduce an importance sampling procedure that ensures that sufficiently many data close to the data support are seen during training. This enables estimating score direction and magnitude at sufficient precision in such environments, which is important for both sharpening image output and accurate termination via tracking estimated score explosion.

For the unconditional generation of MNIST digits we compare our model against locally trained DDPM and DDIM models with U-Net architectures of comparable size to our model implementation. Since our model terminates quickly after $\sim 150$ steps on average, we use the flexibility of the step size schedule for DDIMs to match the average NFEs of our model, while using the DDPM with 1,000 steps as a high-budget baseline.

For the adaptive denoising experiment we show that, as predicted by our theory,

1. the model works very well as a denoiser, when running the pretrained unconditional model on a noisy sample;
2. the model generalises zero-shot to different noise types (Gaussian, Laplace, uniform and Salt \& Pepper) and can therefore be used as a robust denoiser.

We further demonstrate how we can implement the fine-tuning mechanism introduced in Section 5.2 for conditional generation tasks that highlight the adaptive nature of our algorithm by considering inpainting of masked digits and steering the generative process to produce a 3 when initialised in a noisy 8. We also show that the simulation time of the algorithm can indeed be used as a reliable statistic for anomaly detection, and for the concrete model implementation suggest another, computationally more efficient procedure, based on evaluating the one-step projection distance of a given sample.

Finally, as pointed out by one of the reviewers, the suggested classification algorithm from the original manuscript did not quite match the general framework of the paper since it relied on estimates of class supports, which were assumed to be given. We have therefore formulated a new classification algorithm, which works entirely within the conditional generation pipeline and is based on tracking the class corrector energy of conditional scores along an unconditionally simulated path. Empirically, we show that this can serve as an alternative to a simple Denoise-then-Classify algorithm (using our  model as denoiser and an external classifier) that is more robust to the noise level of the data to be classified.

All remaining specific remarks have been addressed in the revised manuscript, see specific answers to the reviewers' comments below. For the convenience of the reviewers, all updates in the revised version are highlighted in blue.